# Spontaneous traveling waves naturally emerge from horizontal fiber time delays and travel through locally asynchronous-irregular states

Zachary W. Davis [1,4✉], Gabriel B. Benigno[2,3,4], Charlee Fletterman[1,4], Theo Desbordes[1], Christopher Steward[3], Terrence J. Sejnowski [1,5], John H. Reynolds [1,5✉] & Lyle Muller [2,3,5✉]

Studies of sensory-evoked neuronal responses often focus on mean spike rates, with fluctuations treated as internally-generated noise. However, fluctuations of spontaneous activity, often organized as traveling waves, shape stimulus-evoked responses and perceptual sensitivity. The mechanisms underlying these waves are unknown. Further, it is unclear whether waves are consistent with the low rate and weakly correlated "asynchronous-irregular" dynamics observed in cortical recordings. Here, we describe a large-scale computational model with topographically-organized connectivity and conduction delays relevant to biological scales. We find that spontaneous traveling waves are a general property of these networks. The traveling waves that occur in the model are sparse, with only a small fraction of neurons participating in any individual wave. Consequently, they do not induce measurable spike correlations and remain consistent with locally asynchronous irregular states. Further, by modulating local network state, they can shape responses to incoming inputs as observed in vivo.

[1] The Salk Institute for Biological Studies, La Jolla, CA, USA. [2] Department of Applied Mathematics, Western University, London, ON, Canada. [3] Brain and Mind Institute, Western University, London, ON, Canada. [4] These authors contributed equally: Zachary W. Davis, Gabriel B. Benigno, Charlee Fletterman. [5] These authors jointly supervised this work: Terrence J. Sejnowski, John H. Reynolds, Lyle Muller. ✉email: zdavis@salk.edu; reynolds@salk.edu; lmuller2@uwo.ca

Visual cortical neurons exhibit variable fluctuations in their spontaneous activity and stimulus-evoked responses. Rather than being due to noise intrinsic to the neural spiking mechanism[1], which is highly reliable[2], variability is thought to emerge from ongoing synaptic activity in the dense recurrent connectivity of cortical networks[3,4]. When observed from a single point in the cortex, spontaneous fluctuations resemble a broadband temporal noise process[4,5]. Multisite recordings have revealed that these temporal fluctuations can be part of waves traveling across a cortical area[6–10]. Spontaneous traveling waves had largely been observed in slow-wave fluctuations associated with anesthesia, sleep, or low arousal[11–13]. While traveling waves had been theorized to have an impact on cortical computation, it was difficult to identify their role since active cortical states exhibit fluctuations that are more complex, dominated by higher-frequency and lower-amplitude activity[14], making these waves harder to detect. Further, driving input is believed to quench variability in ongoing dynamics[15], calling into question the potential impact of traveling waves on evoked activity[16].

Recent work has shown that spontaneous traveling waves are present in the awake state, that they influence the magnitude of sensory-evoked activity, and that—depending on their retinotopic alignment with sensory input—they can improve perceptual sensitivity[17]. However, the mechanisms that generate them, and whether they are consistent with the asynchronous-irregular spiking dynamics characteristic of awake cortex[18], are unknown. Based on their speed of propagation, we hypothesize that these waves result from action potentials propagating along unmyelinated horizontal fibers. To test this hypothesis, we studied a spiking network model across a range of biologically realistic neuronal densities, distance-dependent connection probabilities, excitatory/inhibitory balances, and synaptic conductance states. Importantly, this model incorporated axonal time delays from conduction along unmyelinated horizontal fibers, which shaped ongoing activity patterns into traveling waves consistent with those observed in vivo. Spontaneous traveling waves were apparent in this network model and occurred consistently across a wide parameter range that produced asynchronous-irregular dynamics.

One might wonder whether the occurrence of these traveling waves induces correlated variability, which has been found to impair perception[19]. Results from the spiking network model show this need not be the case. In both the computational model and multielectrode recordings in the marmoset visual system, we found the change in spiking probability due to the wave was low, only sparsely modulating spiking activity. We thus refer to the model as the sparse-wave model and this regime as the sparse-wave regime. This is in contrast to smaller-scale network models where spikes are strongly coupled to the state of traveling waves, producing strong correlations in spiking activity. Rather, at the scale of entire cortical areas, spontaneous waves can emerge in spatially structured shifts in spiking probability and propagate through sparse spiking activity along horizontal fibers, without inducing changes in pairwise correlations in the activity of individual neurons. Traveling waves can thus coexist with a locally asynchronous-irregular state, conferring their benefits while maintaining the computational advantages of this dynamical regime[20,21].

## Results

### Spontaneous synaptic fluctuations are comparable to those during stimulus-evoked responses.
Previous work has shown that moment-by-moment fluctuations in synaptic input in the cortex can be on the same order of magnitude as during the sustained period of stimulus-evoked responses[6,22–24]. Fluctuating

synaptic inputs can have a significant impact on neural excitability[25], gain modulation[26], and readout of sensory information[7]. To understand the impact of the spontaneous network state on evoked responses in the awake visual cortex, we recorded spontaneous and stimulus-evoked activity from chronically implanted multielectrode Utah arrays (Blackrock Microsystems) in area MT of two common marmosets (*Callithrix jacchus*; data previously reported by Davis et al.[17]). Spontaneous multiunit activity recorded from a single electrode while a marmoset fixated a fixation point was characterized by a low, irregular firing rate. The appearance of a highly salient stimulus (10% Michelson contrast drifting Gabor) within the multiunit receptive field evoked a robust response (Fig. 1a). When measured over many repeated presentations of the stimulus, the mean multiunit firing rate rose from $13 \pm 1.6$ sp/s during fixation, to $97 \pm 5.7$ sp/s in response to the stimulus ($N = 40$ trials over three recording sessions). These evoked spiking responses were variable from trial to trial (mean fano factor = $1.01 \pm 0.01$ SEM, 40–240 ms after stimulus onset), consistent with previous observations[15,27,28].

This variability is partly the result of ongoing spontaneous fluctuations in synaptic activity in the local population at the time of the evoked spiking response[6,7]. These fluctuating synaptic inputs, in turn, contribute to the local field potential (LFP)[29,30]. When averaged across high-contrast trials, the LFP had a robust negative deflection aligned to the stimulus-evoked spiking response, while the pre-stimulus period was flat (black line, Fig. 1a). However, at the single-trial level, the stimulus-evoked LFP response was similar in magnitude to the spontaneous fluctuations occurring during fixation (right panel, Fig. 1a). The relative power between the LFP just prior to the stimulus ($-200$ to 0 ms) and following stimulus onset ($+50$ to $+250$ ms) across single trials had a small but significant difference from 0 dB, which represents parity between spontaneous and stimulus-evoked fluctuations (median 1.89 dB, $p = 0.00005$ two-tailed Wilcoxon's rank-sum test).

While strong, high-contrast visual stimulation evoked slightly stronger LFP fluctuations than intrinsic network fluctuations, the distinction disappears in the context of weak visual inputs (Fig. 1b). When the marmoset was presented a faint stimulus that was detected ~50% of the time (<2% Michelson contrast), the evoked spiking response was significantly weaker and more variable (mean = $68 \pm 4.4$ sp/s, $p = 0.0009$; fano factor = $1.54 \pm 0.14$, $p = 0.002$, two-tailed Wilcoxon's rank-sum test). This corresponded with a weaker average LFP response, and the trial-by-trial relative power between spontaneous and evoked fluctuations was not significantly different from 0 dB (median = 1.23 dB, $p = 0.07$ two-tailed Wilcoxon's rank-sum test).

Given the comparable magnitude of spontaneous LFP fluctuations to responses evoked by weak sensory inputs, we hypothesized that much of the variability in neuronal spiking could be explained by the state of the local network since the synaptic drive (manifested in the LFP) during spontaneous and evoked activity is roughly equal[22,23]. We recently reported that spontaneous LFP fluctuations in the awake cortex are organized into waves that travel across an entire cortical area (Fig. 1c and Supplemental Movie S1). They modulate spontaneous spiking probability (Fig. 1d), and they directly impact the magnitude of stimulus-evoked responses depending on their alignment with neuronal receptive fields (Fig. 1e). We found that, rather than acting as a source of noise that impairs perception, spontaneous waves can—depending on their spatiotemporal alignment with a visual stimulus—improve the monkeys' ability to detect the stimulus. We thus sought to understand what mechanisms might generate traveling waves in the cortex and test whether they represent an operating regime either consistent with or distinct

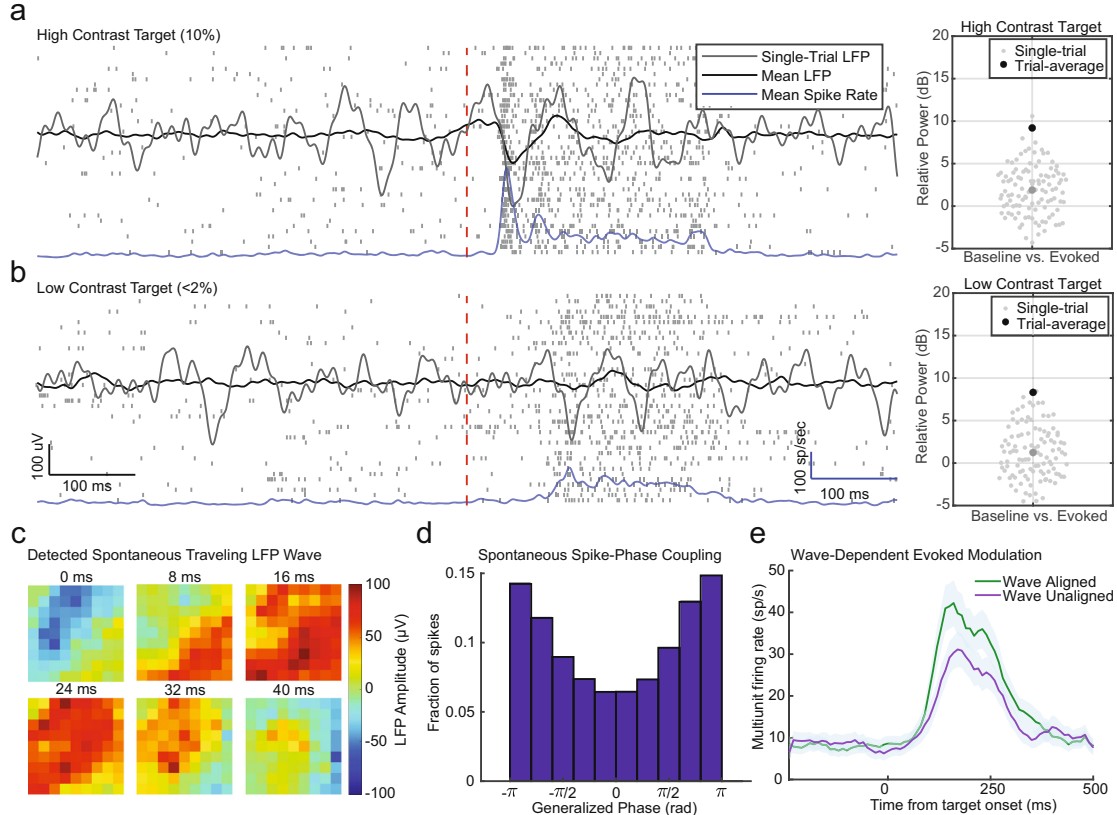

**Fig. 1 Spontaneous network fluctuations are of similar magnitude to stimulus-evoked responses in vivo. a** Spike raster for repeated presentations ($N = 40$) of a high-contrast (10% Michelson contrast) drifting Gabor recorded from area MT of a fixating marmoset (stimulus-onset, red line; mean response, blue line). A single-trial LFP trace is plotted in gray, and the average LFP response is plotted in black. The relative power between baseline ($-200$ ms to stimulus-onset) and evoked fluctuations (stimulus-onset + 50–250 ms) significantly favored the evoked response (right panel; $N = 110$ trials; median = 1.89 dB, $p = 0.000019$, two-tailed Wilcoxon's ranked-sum test). **b** Same as in (**a**), but for a low contrast stimulus (<2% Michelson contrast). The relative power between baseline and evoked LFP fluctuations was not statistically different from parity (median = 1.23 dB, $p = 0.087$ two-tailed Wilcoxon's ranked-sum test). **c** An example of spontaneous LFP fluctuations structured as a traveling wave recorded from a spatially distributed multielectrode array in marmoset area MT. **d** Histogram of spontaneous spike probability as a function of the generalized phase of the LFP during fixation. **e** The average evoked response to low contrast stimuli was stronger when a more excitable phase ($\pm\pi$ rad) of a spontaneous traveling wave aligned with the retinotopic location of the target (aligned, green line) as compared to when a less excitable phase (0 rad) was aligned (unaligned, purple line; $N = 43$ wave and non-wave trials; shaded region SEM; $p = 0.0000015$ two-tailed Wilcoxon's rank-sum test). Data for panels **c**–**e** modified from Davis et al.[17] with permission.

from the irregular, asynchronous activity patterns classically observed in silico[31,32] and in vivo[21,33].

**Spontaneous traveling waves can emerge in network models without altering individual neuron spiking statistics.** To address this question, we studied large-scale spiking network models composed of leaky integrate-and-fire (LIF) neurons with balanced excitation and inhibition and conductance-based synapses. When neuronal interactions are modeled as conductances, taking into account the time-dependent driving forces and channel activations at the synapse, spiking network models can enter into states of self-sustained activity[34,35]. Asynchronous-irregular activity[32] in these self-sustained states, generated without external drive, results naturally from the recurrently generated fluctuations intrinsic to the dynamics of the system[34,35]. These dynamics are characterized by low, variable firing rates, weak pairwise correlations, and coefficient of variation (CV) near unity. These self-sustained states provide an opportunity to study spiking network dynamics that are structured by the recurrent activity of the network itself, rather than dominated by random external Poisson synaptic input[20], and are well suited to model the spontaneous background activity observed in the cortex during active perception.

We first studied a two-dimensional (2D) conductance-based spiking network model with over 1,000,000 neurons distributed over a $6 \times 6$ mm$^2$ area consisting of 80% excitatory and 20% inhibitory neurons, randomly connected with 3000 synapses per cell, yielding a sparsely connected network (Fig. 2a). We eliminated the outer millimeter from analysis, yielding a $4 \times 4$ mm$^2$ area with 450,000 neurons. These values were selected to approximate the density and connectivity of neurons in cortical layer 2/3 of area MT in the common marmoset[36,37]. This randomly connected network generated self-sustained activity with spontaneous spiking fluctuations consistent with the asynchronous-irregular regime[32,38] and lacked any spatiotemporal structure (Fig. 2b, c). A simulated LFP was calculated from summed excitatory and inhibitory synaptic activity over adjacent, nonoverlapping pools of $10 \times 10$ neurons (corresponding to $67.8 \times 67.8$ μm$^2$)[39] and was used to estimate the local excitability state at each point in the network for comparison to the electrophysiological recordings. The LFP was homogeneous across the network, as would be expected from pools of neurons receiving synaptic input from random positions in the network (Fig. 2c and Supplemental Movie S2).

To test whether topographic connections with transmission delays were sufficient to generate spontaneous traveling waves in

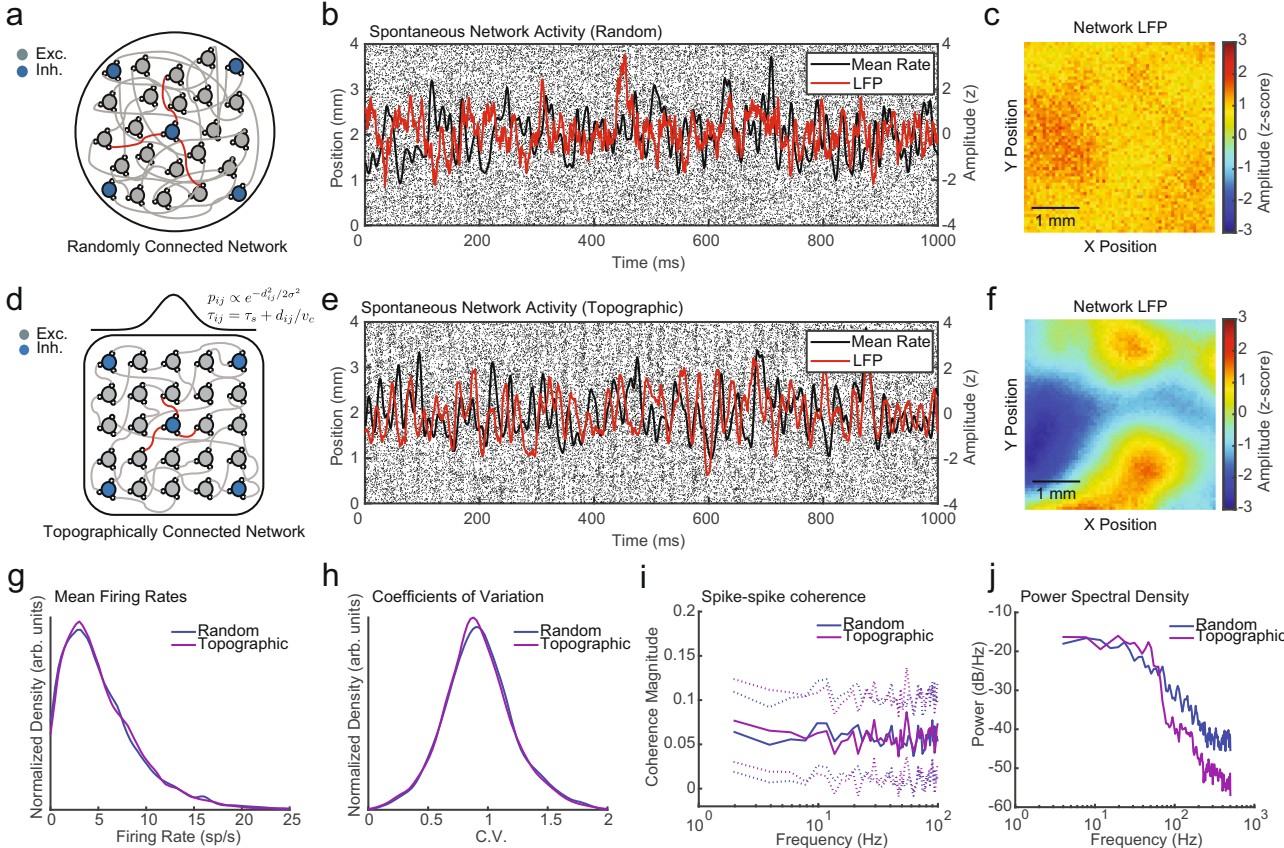

**Fig. 2 Topographically connected networks produce structured fluctuations without altering neuronal spiking dynamics. a** Schematic diagram of a 2D spiking network model with 80% excitatory (gray) and 20% inhibitory (blue) neurons wired with a uniformly random connection probability. **b** Spike rasters from 10,000 excitatory neurons along a 1D slice arranged by linear distance in the network. LFP fluctuations calculated from summed synaptic currents for a single 10 × 10 neuron pool is plotted in red. The mean spike rate within one neuron pool is shown in black. **c** Spatial organization of LFP amplitude for each neuron pool in the network plotted at one time point. **d** Network schematic as in (**a**), but the network was topographically connected with probabilities drawn from a Gaussian ($\sigma = 400\ \mu m$), and activity had a distance-dependent transmission delay (0.2 m/s). **e** Spike rasters as in (**b**), but sparse structured fluctuations were apparent across the network. **f** Spatial LFP amplitude as in (**c**), but the LFP was heterogeneous across the network with topographic structure. **g** The distribution of single-unit mean firing rates did not differ between the random (blue line) and topographic networks (purple line; $N = 5000$ neurons; $p = 0.28$, two-tailed Wilcoxon's rank-sum test). **h** The distribution of single-unit CV did not differ between the random and topographic networks ($p = 0.11$). **i** Pairwise spike coherence did not differ between the random (blue line) and topographic networks (purple line; $N = 10$ paired adjacent neuron pools; CI test, $\alpha = 0.05$; dotted lines 95% CI). **j** Power spectral density for LFP from the random and topographically connected networks.

the network, as a refinement to the model described in Fig. 2a, two key elements were introduced: (1) connection probability decayed as a Gaussian with the distance between neurons ($\sigma = 400\ \mu m$)[40,41] to mimic the topographic connectivity in cortex and (2) action potentials activated synaptic currents after a time delay determined by the distance between neurons to simulate the conduction velocity of horizontal fibers in the cortex ($v_c = 0.2$ m/s;[42] Fig. 2d). This network also produced self-sustained, spontaneous fluctuations, but spiking activity was weakly organized into bands that moved across the network as traveling waves (rasters, Fig. 2e). LFP fluctuations were heterogeneous across the network and exhibited organized spatial structure with localized regions coordinated in amplitude (Fig. 2f and Supplemental Movie S3).

To test whether the presence of these organized topographic fluctuations altered the asynchronous-irregular dynamics of individual neurons in the network, we compared the firing rates and CV across a randomly selected population of excitatory neurons ($N = 5000$). There was no difference in the distribution of firing rates across the networks (mean rate = 5.23 vs. 5.27 sp/s; $p = 0.28$, two-tailed Wilcoxon's rank-sum test; Fig. 2g) or in the

distribution of CV (mean CV = 0.93 vs. 0.92; $p = 0.11$, two-tailed Wilcoxon rank-sum test; Fig. 2h). Therefore, individual neurons maintained their asynchronous and irregular firing states while the topographically connected network produced spontaneous traveling waves.

While one might expect the organized bands of spiking activity would result in increased correlations across neurons, we found no evidence that this was the case. The introduction of topographic connections did not affect pairwise correlations, as the degree of spike–spike coherence between the randomly and topographically connected networks was indistinguishable (Fig. 2i). No change in coherence occurred despite the topographically connected network producing increased power in lower frequencies (30–50 Hz) and reduced power in higher frequencies (>60 Hz) relative to the randomly connected network (Fig. 2j). The spatiotemporal structure could, therefore, exist in these networks without disrupting CV or pairwise coherence because the spiking probability was only weakly modulated by the presence of traveling waves. The probability of a neuron firing a spike at any given millisecond was low, and the peak of a traveling wave only marginally increased spiking probability (2.33%

increase), resulting in only a small fraction of neurons spiking during the peak of any given wave. We, therefore, refer to this as the "sparse-wave" network regime. If it were the case that neurons strongly participated in these fluctuations, then they would show a degree of coherence in the range of frequencies dominated by those fluctuations. To demonstrate this, we simulated a smaller network with fewer neurons and denser connections (model parameters Table S1), which generated spontaneous fluctuations that strongly regulated spiking activity. This "dense-wave" network did strongly modulate spiking activity during traveling waves (26.48% increase in spiking probability), which produced strong spike–spike coherence in the frequency band dominated by fluctuations in the LFP (Fig. S1). This increase in correlation was greatest for nearby locations in the network and was negatively correlated with distance (Pearson's $r = -0.72$; Fig. S2). Thus, unlike in the dense-wave network, traveling waves in the sparse-wave regime do not necessarily induce pairwise correlation across the network.

**Topographic connectivity and distance-dependent delays are necessary to generate spontaneous waves.** As hypothesized, the addition of topographic connections and conduction delays was sufficient to produce clear spatiotemporal organization in the network activity (Fig. 3a). In order to detect traveling waves, we utilized the property that activity patterns propagating at a fixed speed in the network will produce a band at a constant slope in the 2D space–time fast Fourier transform (FFT)[43]. Importantly, although the power spectral density at each point in the network had broad-spectral power (Fig. 2j), the 2D space–time FFT revealed a clear spectral peak (Fig. 3b), whose slope in relation to the temporal and spatial frequencies was dependent on the axonal conduction speed. To classify these activity patterns as traveling waves and quantify their properties relative to cortical recordings from the marmoset cortex, we applied the same analysis technique developed for the experimental recordings (generalized phase, GP[17]) to the simulated LFP in each $10 \times 10$ neuronal pool. We then adapted a technique previously developed for detecting

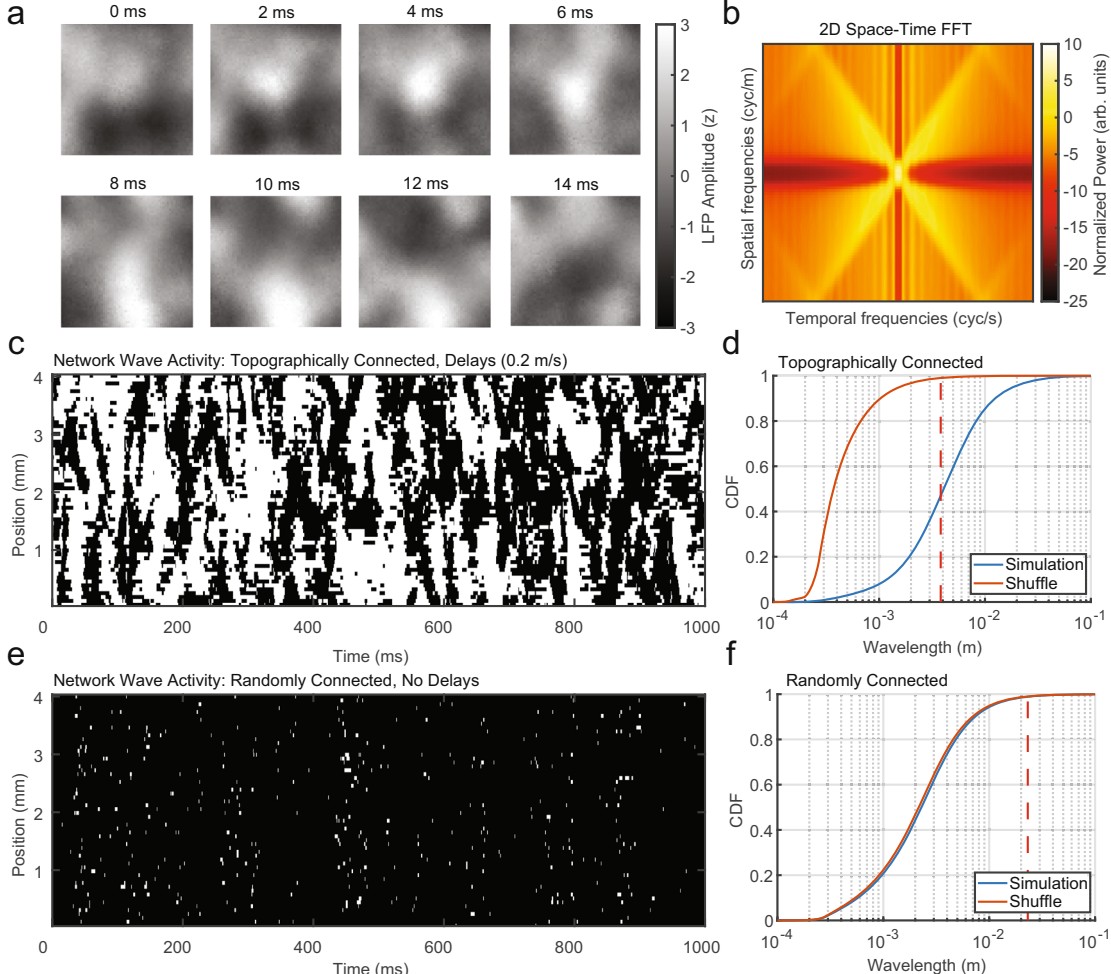

**Fig. 3 Spontaneous topographic network fluctuations travel as waves. a** Time series of simulated LFP activity from the topographically connected network in Fig. 2. Regional peaks and troughs moved coherently across the network as traveling waves. **b** 2D (space–time) FFT reveals a concentration of spatiotemporal energy along temporal (x-axis) and spatial (y-axis) frequencies reflecting the flow of activity across the network. **c** The presence of significant wave activity for a linear slice through the large-scale 2D network model. Significant (white) wave values were defined as estimated wavelengths that exceeded the 99th percentile of the spatially shuffled wavelength distribution (48.53% of network activity). **d** Cumulative distribution functions (CDFs) of the observed wavelengths (blue) and wavelengths after spatially shuffling the LFP pool locations (red, 99th percentile, red dashed line). **e** The randomly connected network had few points that were classified as traveling waves (1.12% of network activity). **f** Wavelength CDFs as in (**d**) for the randomly connected network and its shuffle.

traveling waves in noisy multielectrode recordings[44,45]. We estimated the gradient of the phase at each moment in time and calculated putative wavelengths. We then identified places and times in the network where there was significant spatial organization. Significance was determined by comparing the observed wavelengths to the wavelength distribution after a spatial shuffle of electrode positions, with the 99th percentile of the shuffle distribution taken as the threshold criterion (Fig. 3c, d). This approach provides a sensitive and robust means to detect traveling waves from moment to moment[44,45]. We found significant wave activity in the topographically connected network ~50% of the time, whereas the presence of significantly structured wave organization was absent from networks with random connections and no delays (Fig. 3e) as the distribution of putative wavelengths was similar to the shuffled distribution (Fig. 3f).

We also explored the sufficiency of topographic connections and conduction delays in generating waves separately (Fig. S3). A topographic network lacking transmission delays produced spatially organized activity, but there was no spectral line in the 2D FFT consistent with traveling waves (Fig. S3b). Conversely, delays in an otherwise randomly connected network did not generate large-scale spatially organized activity, but did have a clear spectral line consistent with propagating activity (Fig. S3c). From this, we conclude that, in our framework, topographic connectivity is necessary for the emergence of large-scale spatially organized activity, and transmission delays are necessary for the regular flow of activity over space and time. Both topography and delays together are necessary in our network framework to produce spatiotemporal dynamics that travel over the network consistent with the traveling waves we observed in our cortical recordings. These results were consistent in a simpler one-dimensional (1D) network model where the emergence of traveling waves required both topographic connections and transmission delays (Fig. S4).

**Spontaneous waves occur throughout the asynchronous-irregular regime.** In the example network, topographic connections with axonal conduction delays were sufficient to induce large-scale waves of activity without disrupting the fine-scale asynchronous-irregular dynamics of individual neurons. Does the presence of traveling waves generalize across all networks with asynchronous-irregular activity, topographic connections, and axonal conduction delays[35,38]? We scanned across 2500 combinations of different excitatory and inhibitory (E/I) conductances in the topographically connected model and found 601 combinations that produced self-sustained spiking activity. We then identified networks with asynchronous-irregular spiking dynamics, defined as networks with mean excitatory firing rates between 1 and 25 sp/s and mean CV between 0.7 and 1.4[38]. Approximately 99% (599 out of 601) of the networks that generated self-sustained activity were classified as asynchronous irregular. We then measured the percentage of time each network's activity was significantly organized into traveling waves. Waves were present across the entire range of asynchronous-irregular networks (Fig. 4a). The strength of wave activity was negatively correlated with the magnitude of E/I conductance (Pearson's $r = -0.55 \pm 0.002$, 95% confidence interval (CI)) indicating weaker synapses led to stronger wave activity. The average wavelength was positively correlated with synaptic strength (Pearson's $r = 0.72 \pm 0.001$; Fig. 4b), indicating stronger synaptic weights lead to more synchronous network dynamics. These results demonstrate that spontaneous traveling waves are a general property of topographic connectivity and are entirely consistent with locally asynchronous-irregular states.

How important is network scale in generating traveling waves? To answer this question, we simulated networks ranging from 0.5 to 4 mm in width, holding neuronal and connection density constant. For small networks (~0.5 mm), a very limited range of the E/I space produced self-sustained and asynchronous network dynamics. As network size grew, the asynchronous-irregular parameter space grew as well, extending to include smaller and smaller combinations of E/I synaptic strength[18] (Fig. 4c). It was thus necessary to simulate spiking network models at sufficient spatial scales (>1 mm) to generate asynchronous-irregular activity in networks with conductances resembling those estimated in vivo[18]. At small network scales, wavelength distributions during asynchronous-irregular dynamics were not distinct from the spatial shuffle, and the parameters that favored longer wavelengths did not produce asynchronous-irregular activity. Only at larger network scales did wave activity become strongly apparent (Fig. 4d).

**Network connectivity determines wave properties.** What effect did our chosen parameters for connection distance and conduction velocity have on wave properties? We hypothesized the spatial extent of connections and the conduction speed of spikes directly control the wavelength and propagation speed, respectively, of traveling waves in the model. To test these predictions, we simulated networks with various values of standard deviation ($\sigma_s$) for the Gaussian connection probability distribution. Consistent with our hypotheses, the distribution of significant wavelengths increased with larger connection distances (Fig. 5a and Supplemental Movies S4 and S5), and increasing the conduction velocity created a corresponding increase in the propagation speed reflected in the slope of the spectral line in the space–time FFT (Fig. 5b). Thus, the macroscopic features of spontaneous traveling waves were directly related to specific network structures in the model.

Are these waves only possible with perfectly Gaussian connection profiles and uniform conduction velocity, or can they tolerate a broad range of values similar to those observed in vivo? To test this, we simulated the example model in Fig. 2, with 10% of the connections randomly rewired with uniform probability across the network, generating long-range connections[46,47]. The conduction velocity along each connection was drawn from the range of conduction speeds observed for unmyelinated horizontal fibers in the cortex (0.1–0.6 m/s[42,48,49]). Spontaneous traveling waves persisted under these network conditions (Fig. 5c, d), indicating that the presence of waves was not limited to a fixed or limited set of homogeneous network properties, but instead also occurred in networks with large heterogeneity, as in the cortex.

**Network simulations are consistent with traveling waves in vivo.** How well do the dynamics observed in our simulations match the dynamics observed in electrophysiological recordings of the cortex? To test this, we compared the model results to the data recorded from marmoset MT, while monkeys fixated a spot at the center of an otherwise gray computer screen. We measured the mean firing rates (Fig. 6a) and CV (Fig. 6b) across the population of single- and multiunit activity over multiple recording sessions. The distributions of firing rates and CV were qualitatively similar between the recorded data and the sparse-wave model, suggesting that the spontaneous dynamics in the cortical recordings are also consistent with the asynchronous-irregular regime.

We next measured the distribution of estimated wavelengths in the data and compared this to the wavelength distribution in the model. LFP data in the cortex are not independent across electrodes (as it is in our simulation), but rather pools signals from a cortical volume of ~250 µm in a radius around the electrode tip[29,30] and has correlations that fall off with distance[14].

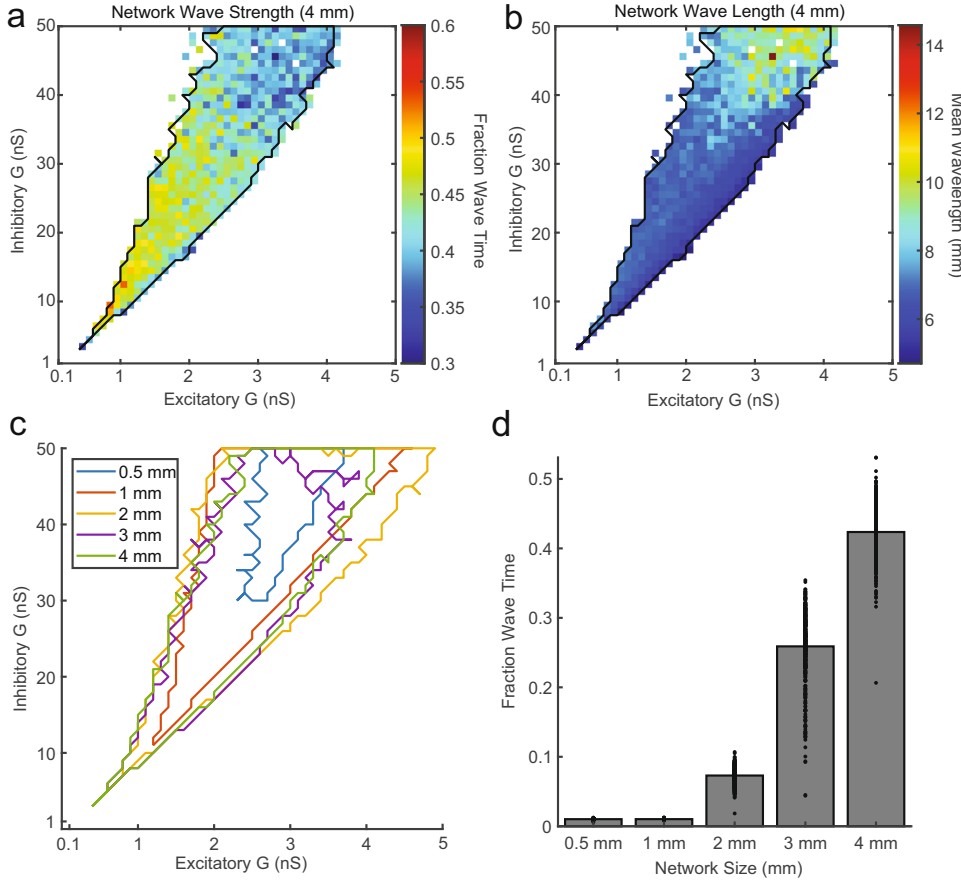

**Fig. 4 Spontaneous traveling waves emerge in network regimes consistent with asynchronous-irregular dynamics. a** The percentage of wave-like activity across ranges of excitatory and inhibitory conductances for 4 mm width topographic network simulations. Each pixel is one simulation of a network as described in Fig. 2d, but with the excitatory and inhibitory conductances corresponding to its x and y coordinate, respectively. Networks consistent with asynchronous-irregular spiking dynamics fall within the black border. White pixels are networks that did not self-sustain or had extremely low/high mean firing rates (FR < 1 or FR > 25 sp/s). **b** Same as (**a**), but the mean of the wavelength distribution is plotted. **c** The region of the parameter space that exhibited asynchronous-irregular activity grew to include smaller synaptic conductances with the size of the network (width 0.5 mm to 4 mm). **d** The fraction of wave activity present across all self-sustained and asynchronous-irregular networks grew with the size of the simulated network (N = 163, 532, 752, 540, and 599 stable A–I simulations for sizes 0.5, 1, 2, 3, and 4 mm respectively).

To emulate these properties of cortical LFP recordings, we applied a smoothing kernel that expanded the area of integration from each simulated LFP point and reduced the independence of the signal. After smoothing our simulated LFP and quantifying wave properties, the distribution of simulated wavelengths closely approximated the distribution observed in the cortex (dotted gray and blue lines, Fig. 6c). Similarly, the distribution of observed speeds in both the network simulation and the data covered the range of conduction velocities in the horizontal fibers, peaking at ~0.2 m/s (Fig. 6d). Thus, across four different measures (spike rate, spike variability, wave size, wave velocity), the distributions characterizing activity in the network model were in close alignment with experimental recordings.

**Neurons sparsely participate in waves due to weak coupling to synaptic fluctuations.** How does activity in the sparse- and dense-wave networks affect the membrane potentials at the level of individual neurons? To answer this question, we studied the membrane potential distributions of individual neurons in each network. In the sparse-wave model, membrane potential fluctuations were Gaussian and close to the spiking threshold, consistent with the fluctuation-driven regime[50] (black line, Fig. 7a). This was in contrast to the skewed distribution of membrane potentials in the dense-wave network, which was consistent with

a synaptic drive to neurons that is clustered and strongly correlated[51] (purple line, Fig. 7a).

In the sparse-wave network model, stochastic fluctuations in the membrane potential produced sparse and irregular spiking activity. These fluctuations were driven by shifts in excitatory-inhibitory balance across the local population, which, due to the topographic network connections, were shared by adjacent populations and carried by spikes across horizontal connections. These summed currents in our estimate of the LFP reflect the total synaptic input in the local population, which exhibited a counter phase relationship with the relative E/I balance: the inhibition-dominated E/I regime produced positive LFP potentials, and the excitation-dominated E/I regime produced negative LFP potentials (Fig. 7b, c). This leads to the mechanistic observation that when the conductances are high, the z-scored LFP is positive and the balance is dominated by the shunting effects of inhibition. When the conductances are low, the z-scored LFP is negative and the balance is shifted to excitation, producing more spiking activity. This relationship mechanistically accounts for the phase-dependent relationship of spiking to the LFP in our cortical recordings.

To demonstrate that simulated neurons are sparsely modulated by traveling waves, we measured the LFP phase at which each spike occurred (10 bins from $-\pi$ to $\pi$; Fig. 7d) across network

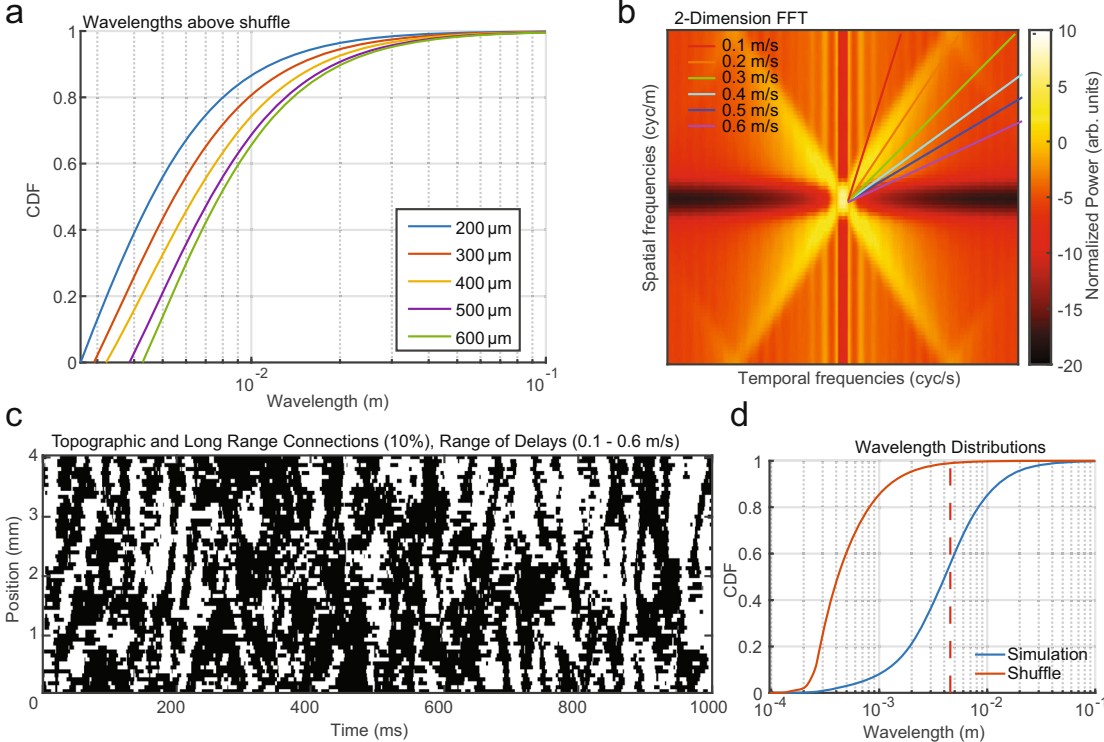

**Fig. 5 Wavelength and speed distributions change with connection distance and conduction velocity parameters. a** Distribution of wavelengths exceeding the 99th percentile of a spatial shuffle for networks simulated across a range of Gaussian sigma values for the topographic connection probability. **b** 2D FFT for the topographic network with delays produced by 0.2 m/s conduction velocities. The colored lines match the slopes of the concentration of spatiotemporal energy across different simulations with conduction velocities ranging from 0.1 to 0.6 m/s. **c** Significant wavelengths (white pixels) were strongly present in a network with a mix of topographically (90%) and randomly (10%) connected projections and a range of conduction velocities (0.1–0.6 m/s). **d** CDFs for the simulation in (**c**) and its shuffle (blue and red, respectively).

simulations with varying E/I synaptic conductances. The degree of spike-phase modulation was significant across the entire parameter space, with spikes more likely during phases closer to ±π. The magnitude of this modulation was correlated with the magnitude of E/I conductances (Pearson's $r = 0.78 ± 0.001$, 95% CI; $N = 599$ simulations), with stronger synaptic weights driving stronger coupling of spiking activity to LFP fluctuations indicating denser and more synchronous spiking waves. This result highlights the importance of large-scale network simulations that can produce stable A–I spiking dynamics with weaker synaptic weights to see sparse modulations of spiking probability by traveling waves. We chose a point among these small conductance values (1 nS $G_e$, 10 nS $G_i$; same values for the topographic network in Fig. 2), to compare the degree of coupling between the model and the cortical recordings. There was no difference between the magnitude of spike-phase modulation observed in the sparse-wave network model and the recorded data ($N = 22$ matched resamplings; model spike-phase index = $0.15 ± 0.001$ SEM; cortex spike-phase index = $0.16 ± 0.005$; $p = 0.18$, two-tailed Wilcoxon's rank-sum test), although the preferred phase-angle differed slightly between the data and model (data-preferred phase = 3.05 rad, model-preferred phase = −2.27 rad).

While there was a similar degree of spike-phase modulation between the cortical data and the sparse-wave model (Fig. 7e, f), the modulation was significantly stronger in the dense-wave model ($N = 10$ resamples; spike-phase index = $0.44 ± 0.01$, $p = 0.000085$, two-tailed Wilcoxon's rank-sum test; Fig. 7g). The phase distribution also differed strongly in peak phase angle (dense-preferred phase = −1.11 rad). In addition, the randomly connected network showed no spike-phase relationship ($N = 22$ resamples; spike-phase index = $0.03 ± 0.001$ SEM, Fig. S5), as

expected from a network where the neurons in the LFP pool draw from inputs distributed throughout the entire network. These results demonstrate that—in the simulated large-scale spiking networks—spatiotemporal organization emerges from weak modulations of spiking probability that produces sparse, phase-modulated spiking activity traveling along topographically distributed horizontal fibers. The presence of a similar spike-phase relationship in vivo, particularly for model conductance states that corresponded to experimental estimates of neuronal conductance states[18], demonstrates that the sparse-wave regime in the model is consistent with the properties of waves observed in the experimental recordings.

**Spontaneous traveling waves modulate responses to inputs.** Finally, we hypothesized the state of network fluctuations in the sparse-wave network model would modulate the magnitude of responses evoked by feed-forward inputs, as previously studied for synaptic noise[4,26] and contextual gain control by visual stimuli[52–55]. To test this, we stimulated one $10 × 10$ neuron pool in the sparse-wave network with a 20 Hz Poisson spike train on 100 afferent synapses to each neuron to mimic feed-forward stimulus input to the network. We stimulated for 10 ms, aligned either to the depolarized or hyperpolarized state of network fluctuations defined, respectively, by the most and least probable phases for spikes to occur according to the network's spontaneous spike-phase distribution. When spiking inputs were aligned to the depolarized state, the evoked spiking responses were boosted (blue lines, Fig. 8a) relative to weaker responses when inputs were aligned to the hyperpolarized state (red lines, Fig. 8a). These effects were consistent with wave-modulated visual responses to motion stimuli observed in area MT in vivo (cf. Fig. 1e). In

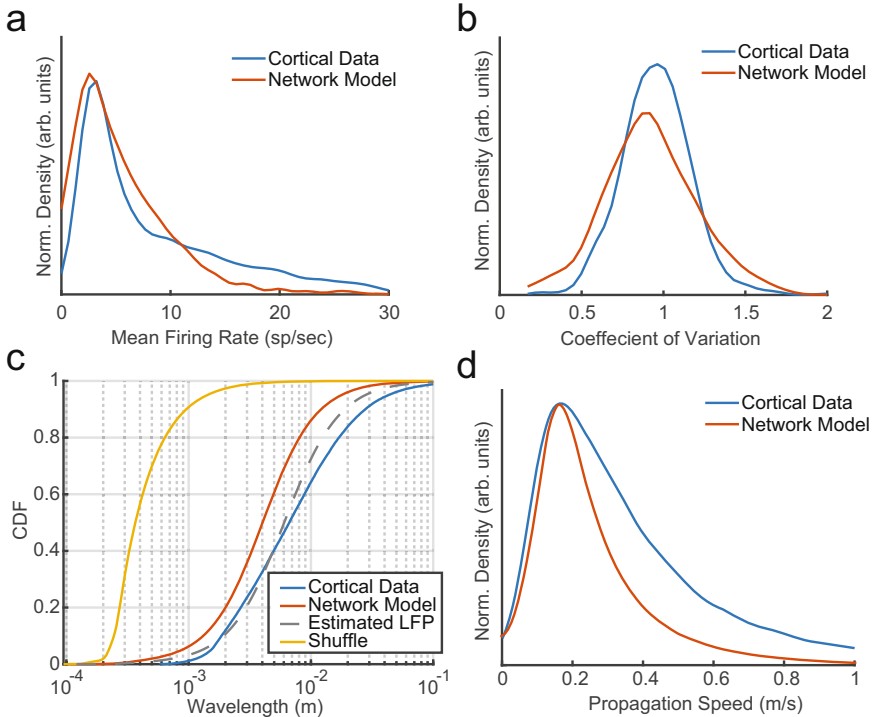

**Fig. 6 Simulated network dynamics approximate dynamics observed in cortical recordings of area MT. a** Distribution of unit mean firing rates across the topographic network simulation in Fig. 2d (red) and the distribution observed across single- and multiunits recorded from area MT of awake, behaving marmosets (blue). **b** The distributions of CV for simulated (red) and recorded cortical data (blue). **c** CDFs of wavelengths for the simulation in Fig. 5c (red), its shuffle (yellow), and the observed distribution recorded in the cortex (blue). Simulated wavelengths were more similar to observed wavelengths in vivo after applying a smoothing kernel across space in the simulated network to approximate the estimated spatial integration of the LFP in electrophysiological recordings (250 μm, gray dashed line). **d** Wave speed distributions for the simulation in Fig. 5c (red), and recorded cortical data (blue) covered a range consistent with axonal conduction velocities (0.1–0.6 m/s).

contrast, equal stimulation in either state of the dense-wave network produced little effect (red and blue lines, Fig. 8b).

To quantify the effect of the traveling waves on evoked responses in each network, we then calculated the gain modulation, which is the ratio of firing rates during the stimulus-evoked response divided by that of the no-stimulus case (Fig. 8c). Across repeated input stimulations, input gain was significantly stronger for the depolarized state relative to the hyperpolarized state in the sparse-wave network ($N = 40$ stimulations; depolarized state $= 3.09 \pm 0.09$; hyperpolarized state $= 2.11 \pm 0.05$, mean ± standard deviation; $p = 3.57 \times 10^{-8}$, two-tailed Wilcoxon's signed-rank test). In contrast, in the dense-wave network, the strong spontaneous fluctuations shunted the incoming spikes, resulting in very weak evoked responses that did not significantly differ depending on network state (depolarized state $= 1.11 \pm 0.05$; hyperpolarized state $= 1.10 \pm 0.04$; $p = 0.20$; two-tailed Wilcoxon's signed-rank test). The increase in gain that occurs in the sparse-wave network mirrors our observations of wave-dependent sensitivity in awake monkeys performing a threshold detection task[17]. Thus, the sparse-wave model offers a mechanistic account for observed phase-dependent modulations of weak sensory responses by traveling waves measured in vivo that a network characterized by dense-wave dynamics fails to replicate.

## Discussion

The present work builds on and seeks to explain our recent finding that spontaneous fluctuations in cortical activity modulate the moment-to-moment processing of sensory information in a manner that affects perceptual sensitivity. These fluctuations are neither synchronous across the cortical surface nor independent noise processes. Rather, they are often structured as traveling

waves. The model presented here shows that distance-dependent conduction delays in topographic, conductance-based spiking network models are sufficient to account for our results in vivo. Waves occur spontaneously, without requiring a driving input, and they occur robustly, in the sense that they are generated across a wide parameter space and in the sense that they occur across the entire space of E/I conductances that gives rise to asynchronous-irregular activity dynamics. The properties of these waves depend systematically on the scales of distance-dependent connections and the speeds of action-potential propagation. The waves were well-matched to those observed in cortical recordings from behaving marmosets[17] for speeds consistent with the conduction velocity of unmyelinated horizontal fibers. Neurons sparsely participated in these waves at the scales of neuronal density and connectivity found in the cortex. The spiking sparseness of the waves allowed them to occur without disturbing the asynchronous-irregular dynamics that are observed in cortical activity and have advantages for neural computation[3,20,21,56,57]. These sparse-wave networks remain sensitive to spiking inputs, producing evoked responses modulated in a phase-dependent manner, as observed in vivo. This is in contrast to smaller-scale networks that exhibit dense waves that drive correlated fluctuations across the population and render the network insensitive to spiking inputs.

These results demonstrate the importance of considering distance-dependent time delays in neural systems. When considered at the scales of entire cortical areas, individual horizontal fibers can span distances ranging from hundreds of microns to several millimeters[46,47], with axonal conduction delays on the order of tens of milliseconds[42,48]. While previous spiking network models that considered relatively smaller spatial scales (from

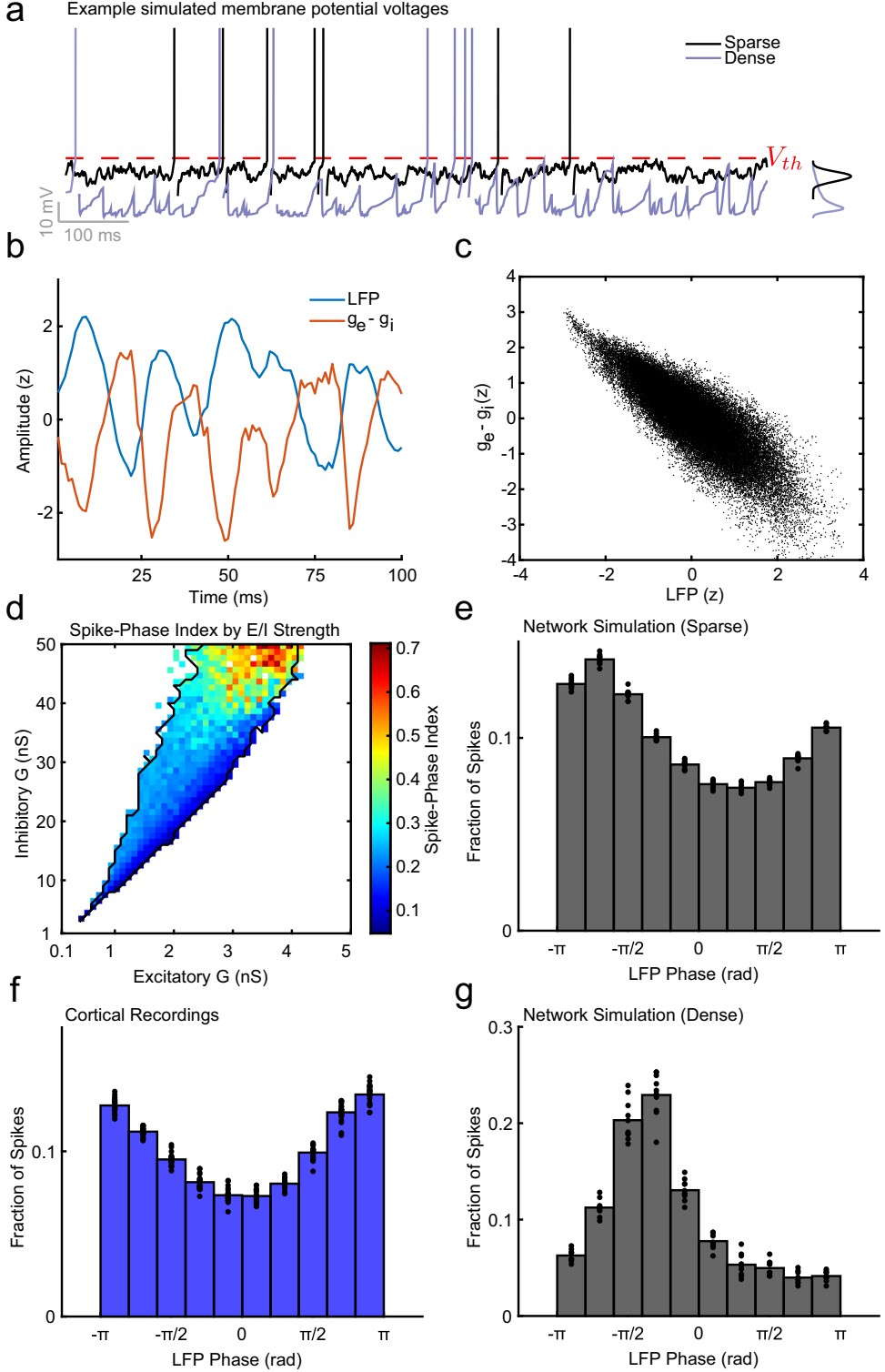

100 μm to 1 mm of cortex) held that the contribution of axonal delays was negligible in their effects on the temporal properties of spiking networks[58], other studies have found fixed delays can have profound effects[59–67]. The effects of fixed and distance-dependent delays have been extensively studied in neural field equations[68–77]; however, in these averaged population models the connection between single-unit and population activity is difficult to study because single-unit information is lost. Finally, traveling waves have been described in smaller-scale topographic spiking network models that lack distance-dependent delays, but these networks only produced dense waves of strongly correlated spiking activity[78–80]. In this work, our large-scale spiking model provides the insight that distance-dependent delays on scales relevant to a large extent of a visual region in the cortex provide a fundamental mechanism to shape spontaneous activity into waves throughout the state of balanced excitation and inhibition. Further, instead of being inconsistent with asynchronous-irregular states, as with previous models of traveling waves in spiking networks, spontaneous waves can travel across these large-scale spiking networks while local networks remain locally

**Fig. 7 Spontaneous waves reflect structured fluctuations in E/I balance that sparsely modulate spike probability. a** Membrane voltage for a simulated neuron in either the sparse-wave network (black line) or dense-wave network (purple line) calculated from the summed excitatory and inhibitory synaptic currents received by that neuron. Spiking activity occurred when the voltage crosses the threshold ($V_{th}$ red line). The distribution of membrane potentials over the interval for the sparse and dense networks is plotted on the right. **b** The amplitude of the simulated LFP (blue line) and the relative level of excitatory and inhibitory conductance (red line) over a 10 × 10 neuron pool were counter phase. **c** Scatter plot of LFP and $g_e - g_i$ difference revealed a significant negative correlation ($N = 50,000$ time points; Pearson's $r = -0.83$; CI test, $\alpha = 0.01$). **d** Spike-phase coupling was significant across networks in the asynchronous-irregular regime, and the degree of coupling was correlated with the magnitude of synaptic conductance ($N = 599$ simulations; Pearson's $r = 0.78 \pm 0.001$, 95% CI). **e** Histogram showing the fraction of spikes that occurred during each phase of the simulated LFP in the topographically connected network. Spike probability was modulated by the LFP phase ($N = 22$ resamples vs. shuffle; spike-phase index = 0.15). **f** Same as in (**e**), but for recorded cortical data. Spike probability was similarly modulated (spike-phase index = 0.16; $N = 22$ recording sessions vs. shuffle). **g** The dense-wave network simulation had a significantly stronger spike-phase relationship ($N = 10$ resamples; spike-phase index = 0.44, $p = 0.0000085$, two-tailed Wilcoxon's rank-sum test).

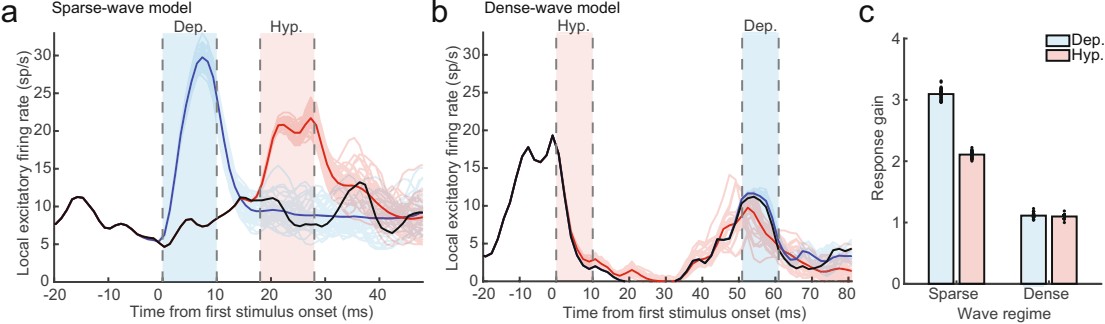

**Fig. 8 The sparse-wave network regime boosts spike inputs while the dense-wave network shunts. a** A 0.2 × 0.2 mm² pool in the sparse-wave network model received a 20 Hz Poisson spike train input for 10 ms aligned either to a period of depolarization (blue shaded region) or hyperpolarization (red shaded region) as defined by the spike-LFP phase relationship. The dark blue and red lines are the mean evoked firing rate after receiving the spiking input in either the depolarized or hyperpolarized phase respectively (light blue and red lines represent $N = 40$ individual trials). The black line is the firing rate of the neuron pool when no input was given. **b** Same as (**a**), but the inputs were delivered to the dense-wave network. The evoked responses were much weaker as the network shunted the currents evoked by the incoming spikes. **c** The response gain between the distributions of spontaneous and evoked activity across $N = 40$ presentations of spiking input. In the sparse-wave network (left bars), inputs during the depolarized state had larger relative responses as compared to inputs during the hyperpolarized state ($3.09 \pm 0.09$ compared to $2.11 \pm 0.05$ mean ± standard deviation; $p = 3.57 \times 10^{-8}$, two-tailed Wilcoxon's signed-rank test). In contrast, the dense-wave network (right bars) responses did not differ in their response gain during the hyperpolarized and depolarized states ($1.10 \pm 0.04$ and $1.11 \pm 0.05$, respectively; $p = 0.20$).

asynchronous. Thus, not only are spontaneous traveling waves consistent with the asynchronous-irregular regime, they are a necessary consequence of topographic connectivity and distance-dependent delays in cortex across conditions that yield asynchronous-irregular spiking.

Critically, the waves we observe sparsely modulate the background spiking probability of neurons in the network, allowing them to maintain locally asynchronous-irregular dynamics. These sparse-wave dynamics only become apparent when networks are modeled at sufficient scales, in particular hundreds of thousands of neurons over an area a few millimeters across. Our custom software implementation allowed for the simulation of networks with ~100,000 to 1,000,000 neurons, each with 3000 outgoing synapses per cell, addressing in the largest networks over 60 GB of RAM. Importantly, while the cells in our network models have a large number of synapses per cell, the number of possible connection partners is high such that the network connectivity remains sparse. In these networks, the large number of synapses per cell allows the network to achieve self-sustained asynchronous-irregular activity when synaptic conductances are on the order observed in cortex[35,81]. Importantly, the sparse waves we observe here may be related to the concept of sparse synchrony[82–84], which has been shown to facilitate information transfer across areas during narrowband oscillations. Sparse waves may reflect this principle unfolding over both space and time, while also being consistent with the more generally

occurring broad-spectral fluctuations during spontaneous awake activity in vivo.

Our findings that traveling waves need not induce pairwise correlations may at first appear to differ from recent work by Huang et al. (2019), in which traveling waves emerged from different spatial and temporal scales of excitation and inhibition and drove shared variability in ongoing dynamics. We do not view these findings as mutually exclusive. The work of Huang et al. offers a mechanistic explanation for a source of shared variability that occurs particularly in low-frequency fluctuations in the sensory cortex[85]. This shared variability has been shown in theoretical[86–88] and experimental studies[19] to have deleterious effects on sensory processing and has been observed experimentally to be reduced by attention[85,89]. In contrast, our model does not generate strong low-frequency dynamics, but instead seeks to describe traveling waves that occupy higher-frequency ranges (above 10 Hz) that our model suggests propagate through horizontal connectivity. Separate mechanisms could underlie the generation of low-frequency correlated variability and higher-frequency traveling waves, the latter of which have recently been shown to have phase-dependent benefits for visual detection[17]. One critical difference between the two models, however, is their relative scale, leaving open the question as to whether the differences in spiking correlation are due to fundamental mechanisms or network size. Additional research is necessary to understand how such mechanisms may interact in large-scale

network models to better recapitulate the broader space of cortical dynamics observed in vivo.

Unlike the outsized conductances typically used in smaller network simulations, the large-scale networks simulated here enabled us to incorporate E/I synaptic strengths that were similar to those observed experimentally, leading to total E/I conductances on the order of the leak conductance[18,90]. This is advantageous because the larger conductances needed in smaller networks yield unrealistic coupling of spiking activity to synaptic fluctuations and shunt driving inputs[91], as illustrated in the dense-wave model. By scaling our model to realistic neuronal densities on a spatial scale over several millimeters of the cortex (450,000 neurons over 16 mm² in the case studied here), the sparse-wave model sustains irregular activity, with strengths of individual synaptic inputs down to 0.5 and 4 nS for excitatory connections and inhibitory connections respectively. At these spatial scales and synaptic conductances, waves are present about 50% of the time, similar to what we find in the neocortex, and the wavelengths closely approximate those we find in vivo.

It is important to recognize that, while our network model of a cortical sheet generates self-sustained activity intrinsically, cortical circuits in vivo are driven by inputs from other cortical areas and subcortical structures, particularly the thalamus. Thalamic inactivation has been shown to severely attenuate the spontaneous firing rates of cortical neurons[92,93], raising the question as to whether spontaneous traveling waves in vivo also involve interactions between cortical and subcortical areas. In this work, our objective was to study whether or not topographic connectivity with conduction delays was sufficient to generate spontaneous waves, as well as to ask whether waves were compatible with asynchronous-irregular dynamics. We, therefore, chose to use the simplest model we could to test what parameters might recapitulate the properties of waves we had observed in the cortex. Undoubtedly, the massive connectivity across cortical areas and subcortical structures impact the features of spontaneous activity in the cortex, and understanding their contribution to the properties of intrinsic traveling waves will be an important avenue of future study.

Traveling waves of neural activity in the awake cortex have been observed under stimulus-evoked[44,94,95], behavior-evoked[96–98], and spontaneous conditions[17]. The fundamental neural circuit mechanism for these waves, however, had remained unclear. Our modeling results suggest that the spontaneous LFP fluctuations we observe traveling as waves in the cortex during active vision result from sparse waves of spiking activity traveling unmyelinated horizontal fibers. The sparse-wave model, which produces activity patterns consistent with the spiking activity observed in vivo, posits that these waves arise from the time delays inherent in communicating spikes across topographic connections within a cortical area. Further, observations from the model suggest that as these sparse waves traverse the massive recurrent connectivity within cortical areas[3,36,99,100], they produce subthreshold shifts in the local E-I balance that, collectively, modulate cortical excitability. Thus, the model offers an explanation for our empirical finding that perceptual sensitivity varies over space and time depending on the alignment of wave phase[17]. Importantly, these traveling waves need not introduce correlated variability believed to harm perceptual sensitivity; instead, the sparse-wave state weakly modulates the background spiking probability in locally asynchronous-irregular neuron pools. Rather than a source of noise, as would be predicted if waves modulated activity akin to the dense-wave regime, the presence of these sparse waves can boost weak inputs that would otherwise have been imperceptible. These results indicate these traveling waves may be a network mechanism that can improve perceptual processing when aligned to the source of feed-forward signals,

without disrupting the computational benefits of the irregular spiking dynamics of individual neurons.

## Methods

**In vivo cortical recordings.** The methods for the recordings and behavioral task used in this work was identical to work previously published[17], which provided the physiology and behavioral data used in this work. Two marmoset monkeys (*C. jacchus*), one male (monkey W) and one female (monkey T), were surgically implanted with a headpost for head stabilization during eye-tracking. The headpost housed an Omnetics connector for a 64-channel multielectrode recording array (Utah array, Blackrock Microsystems), which was implanted in a 7 × 10 mm² craniotomy over area MT (stereotaxic coordinates 2 mm anterior, 12 mm dorsal). Monkey W was implanted with an 8 × 8 recording array with channel spacing of 400 μm and monkey T was implanted with a 9 × 9 array with alternating channels removed yielding a channel spacing of 800 μm. Both arrays had a pitch depth of 1.5 mm. The arrays were chronically implanted over area MT using a pneumatic inserter wand. The craniotomy was closed with Duraseal (Integra Life Sciences, monkey W) or Duragen (Integra Life Sciences, monkey T), and covered with a titanium mesh embedded in dental acrylic. All surgical procedures were performed with the monkeys under general anesthesia in an aseptic environment in compliance with NIH guidelines. All experimental methods were approved by the Institutional Animal Care and Use Committee (IACUC) of the Salk Institute for Biological Studies and conformed to NIH guidelines.

Marmosets entered a custom-built marmoset chair that was placed inside a Faraday box with a liquid crystal display (LCD) monitor (ASUS VG248QE) at a distance of 40 cm. The monitor refresh rate was 100 Hz and gamma corrected with a mean gray luminance of 75 cd/m². Electrode voltages were recorded at 30 kHz from the Utah arrays using two Intan RHD2132 amplifiers connected to an Intan RHD2000 USB interface board. The marmosets were headfixed by a headpost for all recordings. Eye position was measured with an IScan CCD infrared camera sampling eye position at 500 Hz. Stimulus presentation and behavioral control were managed through MonkeyLogic (revision date: 4-05-2014, build 1.0.26) in MATLAB (version R2016b). Digital and analog signals were coordinated through National Instrument DAQ cards (NI PCI6621) and BNC breakout boxes (NI BNC2090A). Neural data were broken into two streams for offline processing of spikes (single- and multiunit activity) and LFPs. Spike data were high-pass filtered at 500 Hz and candidate spike waveforms were defined as exceeding 4 SDs of a sliding 1 s window of ongoing voltage fluctuations. Artifacts were rejected if appearing synchronously (within 0.5 ms) on over a quarter of all recorded channels. Segments of data (1.5 ms) around the time of candidate spikes were selected for spike sorting using principal component analysis through the open-source spike sorting software MClust (ver. 4.3.02; A. David Redish, University of Minnesota) in MATLAB. Sorted units were classified as single- or multiunit and single units were validated by the presence of a clear refractory period in the autocorrelogram. LFP data were low-pass filtered at 300 Hz and down-sampled to 1000 Hz for further analysis.

**Receptive field mapping.** Receptive fields were mapped using a reverse correlation technique. The marmoset was trained to hold fixation on an image (marmoset face, 1 degree of visual angle (DVA) square) presented at the center of the LCD monitor. A drifting Gabor (2° diameter, spatial frequency: 0.5 cycles/degree, temporal frequency 10 cycles/s) appeared at a random position on the monitor between 0° and 18° in azimuth and −15° to 15° in elevation, drifting in one of eight possible directions for 200 ms, after which it disappeared. A new probe then appeared after a random delay drawn from an exponential distribution (mean delay = 40 ms). The sequence repeated until the marmoset broke fixation (defined as an excursion of 1.5° from fixation) or viewed 16 probes. The marmoset was given a juice reward proportional to the number of probes presented. The receptive field for each unit recorded on the array was estimated by calculating the spike-triggered average (STA) stimulus that evoked the maximal response:

$$\mathrm{STA} = \frac{1}{N}\sum_{i=1}^{N} x_i y_i \tag{1}$$

The STA is the sum of probe location $x_i$ weighted by the spike count $y_i$ within the time bin 40–200 ms after probe onset, normalized by the number of all recorded spikes $N$. We estimated the relative position of each recording array in cortex from the location of estimated receptive fields on each spiking channel, and the known topography of area MT in the marmoset[101] (Fig. 1a). We excluded from the analysis the upper half of monkey W's array as the recordings did not appear to be in area MT.

**Behavioral task.** The marmosets were trained to saccade to a marmoset face to initiate a trial of a visual detection task. Upon their gaze landing on the face, the face turned into a fixation point (0.15 DVA). The marmosets held fixation on the fixation point (1.5° tolerance) for a minimum duration (400 ms monkey W, 300 ms monkey T) awaiting the appearance of a drifting Gabor. The Gabor target was 4 DVA in diameter, which reliably produced evoked responses in the multiunit spiking activity on 1–2 adjacent electrodes. The Gabor had a spatial frequency of 0.5 cycles/degree, a temporal frequency of 10 cycles/s, and could drift in one of up

to 8 possible directions. Spontaneous data were analyzed for the period of fixation preceding the appearance of a target and excluded a period of at least 100 ms following the initial saccade to initiate the trial. Early fixation breaks (defined by the excursion of the eye position from the fixation window) were excluded from the analysis. The target only appeared if fixation was held for an additional random duration beyond the minimum duration. The random duration was drawn from an exponential distribution (mean duration = 200 ms) to generate a flat hazard function.

**Relative power between spontaneous and evoked LFP.** We calculated the relative power between spontaneous and evoked LFP (forward-reverse filtered with a fourth-order Butterworth at 5–50 Hz) by computing the sum-squared LFP magnitude in a window onset (0–200 ms) divided by sum-squared LFP magnitude just before stimulus onset (−200 to 0 ms) on the electrode retinotopically aligned to the stimulus location in cortex. For LFP values $\lambda_t$ at this electrode, where $t \in \{\Delta t, 2\Delta t, \ldots, n\Delta t\}$, the relative power $P$ is then

$$P = \frac{\sum_{t_2}^{t_3} \lambda_t^2}{\sum_{t_1}^{t_2} \lambda_t^2} \tag{2}$$

where $t_1 = -200$ ms, $t_2 = 0$, and $t_3 = 200$ ms.

**Computational simulations.** The model consists of $N$ LIF neurons, with $N_e = 0.8N$ excitatory units and $N_i = 0.2N$ inhibitory. The membrane potential $V^{(i)}$ of the $i$th neuron evolves according to the equations

$$C_m \dot{V}^{(i)} = G_L \left( E_L - V^{(i)} \right) + g_e^{(i)} \left( E_e - V^{(i)} \right) + g_i^{(i)} \left( E_i - V^{(i)} \right) \tag{3}$$

$$\tau_{\{e,i\}} \dot{g}_{\{e,i\}}^{(i)} = -g_{\{e,i\}}^{(i)} \tag{4}$$

where $C_m$ is the membrane capacitance, $G_L$ is the leak conductance, $E_L$ is the resting membrane potential, $\tau_{\{e,i\}}$ are the excitatory and inhibitory synaptic time constants, $g_{\{e,i\}}^{((i))}$ are the time-dependent synaptic conductances of the $i$th neuron, and $E_{\{e,i\}}$ are the reversal potentials for excitatory and inhibitory synaptic transmission, respectively.

When $V^{(i)}$ exceeds threshold $V_T$ at time $t_s$, the following spike and reset conditions occur:

$$V^{(i)} \mapsto V_r \tag{5}$$

$$t_n = t_s + \tau^{(i,j)}, \ g_{\{e,i\}}^{(j)} \mapsto g_{\{e,i\}}^{(j)} + G_{\{e,i\}} \ \forall j \in [1, K] \tag{6}$$

where $V_r$ is the reset potential, $t_n$ is the time at which the postsynaptic neuron receives its input following axonal and synaptic delays, $G_{\{e,i\}}$ are the excitatory and inhibitory synaptic weights, $g_{\{e,i\}}^{((j))}$ are the excitatory and inhibitory conductances of postsynaptic neuron $j \neq i$, respectively, and $K$ is the number of postsynaptic targets of neuron $i$. Immediately after neuron $i$ spikes, it undergoes a refractory period of $\tau_r$ where the membrane potential is not updated.

**Network connectivity and axonal conduction delays.** We studied spiking network models with unstructured, random connectivity (random networks, Fig. 2a) or topographic, locally random connectivity (topographic networks, Fig. 2d) or a dense version of the topographic network (dense network, Figs. 7 and 8). The $N_e = 0.8N$ excitatory neurons, of indices 1, 2, …, $N_e$, where $N_e$ is a square number, are arranged uniformly on a 2D grid. Similarly, the $N_i = 0.2N$ inhibitory neurons, of indices $N_{e+1}, N_{e+2}, \ldots, N$, where $N_i$ is also a square number, are arranged uniformly on a 2D grid. Both grids have side length $L$ and they are concentric, together forming a 2D sheet of the $N$ neurons.

In the random network, connections were randomly and uniformly drawn, and the only delay modeled was that relating to synaptic vesicle release, $\tau_s$, which was short and homogeneous across the network. In the topographic and dense networks, connections were randomly drawn from an isotropic 2D Gaussian probability distribution of zero mean and SD $\sigma$ in either dimension. $\sigma$ is 400 μm except in Fig. 6a, where the effect of this parameter was studied systematically. In all networks, there were no self- or double-connections. Axonal conduction delays increased linearly with distance between pre- and postsynaptic cells:

$$\tau^{(i,j)} = \tau_s + \frac{d^{(i,j)}}{v_c^{(i,j)}} \tag{7}$$

where $\tau^{(i,j)}$ is the delay from neuron $i$ to neuron $j$, $\tau_s$ is the same delay representing synaptic vesicle release as in the random network, $d^{(i,j)}$ is the Euclidean distance between neurons $i$ and $j$, and $v_c^{((i,j))}$ is the axonal conduction speed for the connection from neuron $i$ to neuron $j$. All distances were calculated taking 2D periodic boundary conditions into account, effectively wrapping the network onto a toroidal topology[58,81]. 1D versions of the random and topographic networks were also simulated. The models were the same as in the 2D cases, except the neurons were positioned on a ring of length L with periodic boundary conditions.

**Self-sustained activity.** Instead of initializing self-sustained activity through a "kick" of external Poisson input spikes[34,35,38], which may induce trace activity correlations, we recorded the state variables of a self-sustained network, including membrane potential ($V^{(i)}$) and conductance ($g_{\{e,i\}}^{((i))}$), after a long period (10 s) of simulated self-sustained activity. Taking these distributions as a steady state, we then used the Gaussian approximation (mean and variance) to initialize the membrane potentials and conductances with randomly drawn values in the simulations thereafter. After starting the simulation with these initial conditions, networks with approximately balanced excitation and inhibition exhibit self-sustained, irregular spiking activity. Each simulation ran for 1.2 s, from which we eliminated the first 200 ms from our analysis in case of residual initialization artifacts.

**Spike train statistics and the asynchronous-irregular regime.** To characterize basic spike train statistics, we randomly selected 5000 neurons in the simulation and measured the mean firing rate, CV (defined as the ratio of the standard variation of the interspike interval to the mean for each neuron that has a minimum of three spikes over the simulation window), and the average pairwise correlation (average Pearson's correlation between spike trains smoothed with a 100 ms window for 1000 randomly selected pairs). To prevent longer simulations with high firing rates during our parallel runs, networks that produced mean firing rates over 25 sp/s had an early exit condition. Simulations were classified as asynchronous irregular if the mean firing rate across all simulated units was >1 and <25 sp/s; the mean CV across all units was >0.7 and <1.4[38,102].

**Pairwise spike coherence.** Pairwise spike coherence was calculated using multi-taper methods[85]. We took the spike trains from the $10 \times 10$ excitatory neurons comprising the pool for estimating the LFP and an adjacent LFP pool. The 1000 ms of simulation time was broken into 500 ms epochs, stepping 125 ms to cover the full period. The DC component of each unit's spike train was removed, and tapered with a single Slepian taper, giving an effective smoothing of 2.5 Hz for the 500 ms data windows.

To estimate the coherence between two spike trains $x = [x_1 \ x_2 \ldots x_i \ldots x_n]$ and $y = [y_1 \ y_2 \ldots y_i \ldots y_n]$, we first calculated their FFT spectra $X = [X_1 \ X_2 \ldots X_j \ldots X_m]$ and $Y = [Y_1 \ Y_2 \ldots Y_j \ldots Y_m]$, respectively, where $j$ denotes the index of spectral frequency. The auto- and cross-spectral densities are calculated as

$$S_j^{xx} = \frac{2\Delta t^2}{T} X_j X_j^* \tag{8}$$

and

$$S_j^{xy} = \frac{2\Delta t^2}{T} X_j Y_j^* \tag{9}$$

respectively, where $\Delta t$ is the sampling interval, $T$ is the spike train duration, and superscript * denotes complex conjugation. In practice, $x$ and $y$ each represent pools of 100 concurrent spike trains across space. The coherence at a given spectral frequency is calculated as

$$C_j^{xy} = \frac{\left| S_j^{xy} \right|}{\sqrt{S_j^{xx} S_j^{yy}}} \tag{10}$$

This coherence calculation is averaged across ten trials to generate an estimate of the average coherence at each frequency as well as an estimate of the variance. For estimating differences in pairwise coherence across networks, we take the frequency with the maximum coherence in the two networks.

**NETSIM software.** Simulations were generated using a specialized program called NETSIM (v0.1), which is ~1500 lines of C code (available at http://mullerlab.github.io). Equations in the model were integrated using the forward Euler method with a time step of 0.1 ms. Simulation results were additionally point-checked with shorter time-steps throughout. Random numbers were generated using a C implementation of the ISAAC algorithm[103] (Tom Bartol, personal communication, 2016). To verify the numerical integration in this program, we confirmed the network displayed the correct firing rate for unconnected LIF neurons with varying DC-current injections. We also verified simulations under a simple feed-forward network topology to confirm the accuracy of the simulations. In addition, to ensure reproducibility of our computational simulations[104], we compared results from NETSIM and Brian2 at specific points in the $(G_e, G_i)$ parameter space for the balanced random network model, verifying that the mean firing rate, CV, and cross-correlation were in agreement between the two simulators.

**Network parameter scans.** In order to identify the excitatory and inhibitory synaptic conductance weights that produced self-sustained and asynchronous-irregular activity, we simulated networks with 50 values of $G_e$ ranging from 0.1 to 5 nS and $G_i$ ranging from 1 to 50 nS for a total of 2500 simulations (Table S2). In order to determine the effect of network scale on the range of these 2500 simulations that produced self-sustained and asynchronous-irregular activity, we repeated these simulations five times with varying parameters of network size, neuron number, and connections per neuron (Table S2). The number of neurons per

network size was chosen to maintain a density of 28,125 neurons/mm$^2$. The number of connections was chosen to maintain the density of connections within the Gaussian used to assign connections across network sizes. For larger networks (> 2 mm), the connection number did not grow with the size of the network, as 99% of the connections occur within 3 SDs of the Gaussian ($\sigma = 400\,\mu$m). In order to run these simulations across all combinations of network size and conductance parameters, we utilized the Extreme Science Engineering Discovery Environment Comet cluster at the San Diego SuperComputer center at UC San Diego. Data analysis for these simulations was also performed on the Comet cluster running MATLAB. Circular variables were analyzed using the Circular Statistics Toolbox.

**Calculation of LFP estimate**. We utilized a previously developed proxy for the LFP generated by a network of point LIF neurons, which was systematically developed from a spatially extended model[39]. The LFP estimate $\lambda(t)$ is computed as a weighted sum of the excitatory and inhibitory synaptic currents $I_e$ and $I_i$ across $m$ excitatory cells in each $10 \times 10$ neuron pool:

$$\lambda(t) = \sum_{j=1}^{m} I_e^{(j)}(t - \tau) - \alpha \sum_{j=1}^{m} I_i^{(j)}(t) \tag{11}$$

where $\tau = 6$ ms, $\alpha = 1.65$, and $m = 100$ excitatory cells. These values of $\tau$ and $\alpha$ were found by the authors to have an optimal agreement with the LFP generated from a three-dimensional model of spatially extended multi-compartment model neurons[39] and are the values used here. Here, we computed the LFP using the pooled excitatory and inhibitory synaptic conductances and the driving force between the mean pooled membrane potential and the synaptic reversal potential to calculate the average current in the pool. We verified that this approach was nearly precisely equivalent to the proxy calculated using synaptic inputs to each individual neuron in the pool. This LFP proxy is then computed for each $10 \times 10$ neuron pool across the 2D network. The LFP proxy was thus independent across each spatial pool, unlike cortical recordings where LFP signals show varying frequency-dependent scales of spatial integration[105]. We note that excluding these effects is a conservative step, as the addition of spatial integration would only increase traveling waves in the LFP. Further, we note that our results do not depend critically on the choice of LFP proxy and our conclusions are unchanged when analyzing the mean membrane potentials or excitatory synaptic conductances.

In order to compare the properties of waves in our model, where LFP signals are independent across space, with waves recorded from the cortex, where electrodes pool signals across a volume ~250 μm in radius[29,106], we convolved the LFP with a 2D Gaussian kernel (with a spatial standard deviation of four LFP bins, corresponding to a radius of 272 μm) before further analysis.

**Analysis of spatiotemporal dynamics**. To analyze spatiotemporal dynamics in the population activity produced by the spiking network model, we used a technique we recently developed for the wideband analysis of nonstationary data. Briefly, for each real-valued time series $\lambda_{(x,y)}(t) \, \forall x \in [1, N_c], y \in [1, N_r]$, where $N_c$ and $N_r$ are the numbers of columns and rows, respectively, we compute the GP $\phi_{(x,y)}$ of the wideband filtered LFP (fourth-order Butterworth from 5 to 100 Hz) at each point using the corrected analytic signal representation introduced in recent work[17]. We next calculated the gradient of GP $g_{(x,y)}(t)$ at each moment in time:

$$g_{x,y}(t) = -\nabla \phi_{x,y}(t) \tag{12}$$

For the spatial gradient, derivatives are taken across the two dimensions of space and are approximated by the appropriate forward and centered finite differences (formulas and code available in the wave MATLAB toolbox: https://github.com/mullerlab/wave-MATLAB/blob/master/analysis/phase_gradient_complex_multiplication.m). As in previous work, phase differences were implemented as multiplications in the complex plane[44,107],

$$\Delta \phi_n = \arg \left[ \Lambda_{n+1} \Lambda_n^* \right] \tag{13}$$

so that the unwrapping phase across the two dimensions of the network was not necessary. Here, $\Lambda$ is the analytic signal representation of $\lambda(t)$. Wavelength is the reciprocal of the phase gradient magnitude at each point in space and time:

$$\nu_{x,y} = \frac{1}{|g_{x,y}|} \tag{14}$$

As specified in the main text, significance was determined at each point in space and time by comparing observed wavelengths to a spatial shuffle of electrode positions, with the 99th percentile of the shuffle distribution serving as a threshold. The fraction of wave state (Fig. 4a) is the ratio between points with detected waves over all points $\alpha_{w/\alpha}$, where $\alpha_w$ is the number of points with detected waves and $\alpha$ is the total number of points tested.

Wave speed $s(t)$ was computed as the ratio of instantaneous frequency to phase gradient magnitude[96],

$$s(t) = \frac{\frac{\partial \phi}{\partial t}}{|g_{x,y}|} \tag{15}$$

We further analyzed the spatiotemporal activity patterns using a 2D spectral decomposition in space and time (Figs. 3b and 5b). To do this, we calculated the

2D FFT of $\lambda_{(x,y)}(t)$ for each 1D slice through the network by transforming first in space, and then in time. To account for the spatial and temporal autocorrelation in the data, each slice's spectrum was normalized by dividing the spectrum produced from a spatial and temporal shuffle respectively. This normalization allows visualization of the spectral line representing traveling waves in the network LFP; it is important to note, however, that the spectral peak representing traveling waves is nevertheless clear in the raw spectrum. The normalized spectrum for each slice through the network was then averaged together.

**Calculation of response gain**. To quantify the sensitivity of the sparse- and dense-wave network regimes to incoming stimulation, we first identified depolarized and hyperpolarized states from the LFP of a $0.2 \times 0.2$ mm$^2$ neuron pool defined by the spike-phase bins that generated the maximum or minimum spiking probability, respectively, for each network regime. We then applied feed-forward stimulation of 20-Hz Poisson spiking inputs to 100 synapses for each neuron within the pool for 10 ms, aligned to the depolarized or hyperpolarized phase in the network. This process was repeated across 40 trials, yielding a distribution of evoked responses. The same random seed was used to construct the networks across each trial, so that the simulations were identical up to the point of stimulation. We calculated the sum of firing rate during stimulus for the evoked response divided by that of the no-stimulus case.

**Calculation of the spike-phase index**. The degree of spike-phase coupling was measured as the mean resultant vector length for the LFP (filtered with a forward-reverse fourth-order Butterworth filter from 5 to 100 Hz) phase distribution from observed spike times. This measure was calculated using the circ_r function in the Circular Statistics Toolbox for MATLAB (P. Berens, CircStat: A MATLAB Toolbox for Circular Statistics, Journal of Statistical Software, Volume 31, Issue 10, 2009). The mean resultant $r$ of the spike-phase distribution is the normalized sum over complex exponentials of the phase angles $\phi_j$,

$$r = \frac{1}{M} \sum_{j=1}^{M} e^{i\phi_j} \tag{16}$$

where $M$ is the number of spikes, the modulus of $r$ ($|r| \in [0,1]$) represents the degree of spike-phase modulation, and $i^2 = -1$. The closer $r$ is to 0, the more uniform the phase distribution. The closer it is to 1, the more concentrated the phases.

**Statistics and reproducibility**. Experimental results from in vivo electrophysiology were generated in an initial monkey and replicated in a second monkey with a similar result. All analyses that stemmed from previous experimental work were reproduced from newly written analysis code. Network simulations and subsequent data analysis including statistical tests were initially generated and then repeated on separate machines across different institutes to ensure the reproducibility of the results.

**Reporting summary**. Further information on research design is available in the Nature Research Reporting Summary linked to this article.

## Data availability
Access to the raw simulation data and the processed electrophysiology data used in this study are available at https://github.com/mullerlab/davis2021ncomms. Source data are provided with this paper.

## Code availability
An open-source code repository for all analysis methods is available on https://github.com/mullerlab/davis2021ncomms. The open-source code for the custom simulation framework NETSIM is available at https://github.com/mullerlab/NETSIM.

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

## Acknowledgements

We thank Katie Williams, Sean Adams, and Mat LeBlanc for their contributions to this project, in addition to the San Diego Supercomputer Center for their support for this project. We would also like to thank Tom Bartol for the discussions on the software implementation. This work was funded by Gatsby Charitable Foundation, the Fiona and Sanjay Jha Chair in Neuroscience, the Canadian Institute for Health Research and NSF (NeuroNex Grant No. 2015276), the Swartz Foundation, NIH Grants R01-EY028723, T32 EY020503-06, T32 MH020002-16A, P30 EY019005, and BrainsCAN at Western University through the Canada First Research Excellence Fund (CFREF).

## Author contributions

Conceptualization: Z.W.D., G.B.B., C.F., T.D., T.J.S., J.H.R., L.M.; data curation: Z.W.D., G.B.B. C.F., T.D., C.S., L.M.; formal analysis: Z.W.D., G.B.B., C.F., T.D., L.M.; funding acquisition: Z.W.D., T.J.S., J.H.R., L.M.; investigation: Z.W.D., G.B.B., L.M.; methodology: Z.W.D., G.B.B., C.F., T.S., C.S., J.H.R., L.M.; supervision: T.J.S., J.H.R., L.M.; visualization: Z.W.D., G.B.B., L.M.; writing—original draft, Z.W.D., G.B.B., C.F., T.D., J.H.R., L.M.; writing—review and editing, Z.W.D., G.B.B., T.J.S., J.H.R., L.M.

## Competing interests

The authors declare no competing interests.
