## [Peer Review File · Nature Communications]

Spontaneous traveling waves naturally emerge from horizontal fiber time delays and travel through locally asynchronous-irregular statesREVIEWER COMMENTS

Reviewer #1 (Remarks to the Author):

This is an interesting paper in which the authors simulate a large population of LIF neurons that are connected in a topological manner with distance dependent axonal delays. They aim to show that they can maintain irregular activity while at the same time generate traveling waves in the LFP. Finally they show that during certain phases of the wave, there is greater sensitivity to stimuli. The motive of the paper was an earlier paper by the same authors that showed such behavior in marmosets. I have a number of questions about the model and the interpretation. I have no major points but many minor ones that may take some work.

--- Page 6 ---

Authors say:

"While one might expect the organized bands of spiking activity would result in increased correlations across neurons, we found no evidence that this was the case. The introduction of topographic connections"

This seems to contradict the recent paper by Huang et al (Huang, Chengcheng, et al. "Circuit models of low-dimensional shared variability in cortical networks." *Neuron* 101.2 (2019): 337-348) which showed a low dimensional shared variability both in experiments and in simulations. Do the authors have some idea why this differs from the present work? I believe their network was also sparsely connected as it was in the balanced regime and there were wavelike patterns seen as well.

It is hard to believe that there is no correlation for nearby neurons as a wave passes by or is it just that there was such a low probability of a given neurons being involved in the wave?

However the difference between sparse and dense is interesting. The authors should plot the pairwise correlations of nearby neurons as a function of density of connection. I'd be interested in how this varies and at which point it is nonzero.

--- Page 10 ---

"sparsely modulated"

What does sparsely modulated mean?

--- Page 12 ---

Authors say:

"While previous computational models held that the contribution of axonal delays was negligible in their effects on the temporal properties of spiking networks,"

There are a number of older papers that take distance dependent delays into consideration with regard to waves both in neural oscillations and waves. IN all of these the delays matter and lead to dynamics such as traveling waves among others. These should perhaps be discussed. Here is a list (not comprehensive)

Crook, Sharon M., et al. "The role of axonal delay in the synchronization of networks of coupled cortical oscillators." *Journal of computational neuroscience* 4.2 (1997): 161-172.

Atay, Fatihcan M., and Axel Hutt. "Neural fields with distributed transmission speeds and long-range feedback delays." *SIAM Journal on Applied Dynamical Systems* 5.4 (2006): 670-698.

Golomb, David, and G. Bard Ermentrout. "Continuous and lurching traveling pulses in neuronal

networks with delay and spatially decaying connectivity." Proceedings of the National Academy of Sciences 96.23 (1999): 13480-13485.

Golomb, David, and G. Bard Ermentrout. "Effects of delay on the type and velocity of travelling pulses in neuronal networks with spatially decaying connectivity." Network: Computation in Neural Systems 11.3 (2000): 221-246.

Venkov, N. A., Coombes, S., & Matthews, P. C. (2007). Dynamic instabilities in scalar neural field equations with space-dependent delays. Physica D: Nonlinear Phenomena, 232(1), 1-15.

--- Page 15 ---

Authors say:

"Spontaneous LFP fluctuations recorded simultaneously from a spatially distributed multielectrode array were structured as traveling waves"

Note (yellow):

I don't see any traveling wave here. I see a transient increase in amplitude but nothing that looks like an actual moving wave. Am I missing something? Does the fact that the waves are "traveling" change the histogram shape in a way that would be different if these were just synchronized oscillations? That is, I see that the response is stronger at certain phases of the lfp, but I'd expect the same if no traveling was involved. I don't see the role of the motion of the waves in the enhancement of the response. Indeed, it is hard to see how such motion could be relevant to a non moving stimulus. As an example of traveling waves having a computational role to a perception of a stimulus, see the recent paper of Heitmann and Ermentrout in PLoS Comp Bio 2020

--- Page 17 ---

The authors seem unaware of the work of the Doiron group on spatially balanced networks. See the recent paper with Huang et al (op cit) fig 3 and 5 for examples of broad traveling waves. See also the papers with Rosenbaum and Doiron in PRX 2014 and Nature Neuroscience 2017

Have you looked at a simpler one d system and the appropriate mean field spatial model?

--- Page 18 ---

It may be that the reason that you need axonal delays to get the traveling waves is that you assumed that the spatial spread of excitation and inhibition were the same. As the paper does not specify in the parameter tables the time scales of the e and i synapses I could not say if this was also a reason. As seen in the Huang et al paper, it is easy to get wavelike activity without axonal delays when the spatial and temporal time scales are in the right ranges.

I am puzzled by what is plotted in 3c. The network is two d and what is plotted is one d. Is this a slice or what is the spatial position denoting?

If the connectivity is random but there are distance dependent delays will the waves appear? If there is topological connectivity but no delays do the waves appear? I only saw simulations that had both features and I'd like to see which are necessary. That is, if the amplitude coupling is independent of distance but the delay does depend on distance matters more than if the delay is fixed and the amplitude varies with distance. It is known (say in the Crook et al paper above) that distance-dependent delays can destabilize the synchronous state. Is that what is happening here? This is why a mean-field and/or 1-dim network would be useful. It is not clear what the underlying mechanism is.

--- Page 19 ---

The smallest wavelength is 5 mm with the average about 6.5 mm so that there isn't even one wave in the whole network, right since it is only 4 or 6 mm?

--- Page 22 ---

7c is just reflecting your definition in which the LFP is just the difference between g_i and g_e so it should be strongly anticorrelated. Am I missing some bigger point?

--- Page 24 ---

Fig 8 is the most interesting part of the paper.

--- Page 33 ---

In the LFP proxy, where do the values of the parameters come from? That is why is $\tau = 6$ and $\alpha = 1.65$?

--- Page 36 ---

"Ermentrout, G. B." is written twice

Reviewer #2 (Remarks to the Author):

The paper presents an extensive simulation study of a large-scale spiking neural network model with the aim to investigate the mechanisms of traveling waves in cortex reported in their previous paper. A mechanistic understanding of cortical traveling waves is of critical importance for understanding the operating regime of cortex. The authors find that distance-dependent transmission delays and connection probabilities in a network of conductance-based integrate-and-fire neurons are sufficient to reproduce the wave activity. They uncover a new regime, where waves co-exist with asynchronous irregular dynamics typical for cortex. In particular, this regime exhibits a flat spike-spike coherence of low magnitude and wave patterns at the same time. The characterization of the waves relies on their previously developed method and a comparison of the wave length' to wave length' extracted from a corresponding random reference network. They call this new regime a sparse-wave regime. Importantly, the results are compared to experimental data from their previous paper. It is shown that both the spontaneous and evoked activity in this sparse-wave regime is in impressive quantitative agreement with the experimental data. Thus, the major conclusion is that cortical waves operate in the sparse-wave regime. I found the results of the paper very interesting, highly relevant and novel, especially the sparse-wave regime. The simulation and the analysis of simulation data has been performed thoroughly and seems to be technically sound. I think the paper will inspire both experimentalists and theoreticians in neuroscience. My only major concern is the proper characterization of the sparse-wave regime and the computation of the spike-spike coherence. Addressing this issue is important because the sparse-wave regime seems to be the main novelty of the paper. Below I also provide a list of minor issues. In summary, I recommend publication in Nature Communications after these issues have been properly addressed in a revised manuscript.

Major issues

=====

1) The main phenomenon described in this paper is the sparse-wave regime. However, a clear mathematical definition or general conditions for classifying traveling waves as sparse or dense are missing.

1a) Sparse waves are roughly defined by the authors as waves "with only a small fraction of neurons spiking in any given wave". This definition should be made mathematically precise by defining a measure that directly quantifies the notion of the "fraction of neurons spiking in any given wave". E.g one could compute the mean number of spikes "per wave" and neuron, maybe by computing the ratio of mean interspike interval to mean wave duration (at some local position). This direct measure should be applied to network simulations to convince the reader that the fraction of neurons spiking per wave is indeed much smaller than unity for the presumed sparse-wave network, whereas such a property would not hold for the presumed dense-wave network.

1b) As an indirect way to quantify sparse waves the authors consider the spike-spike coherence and argue that "If it were the case that neurons strongly participated in these fluctuations, then they would show a degree of coherence in the range of frequencies dominated by those fluctuations." While I very much like this idea to use the coherence to characterize sparse waves, I have some concerns about the specific procedure used in the paper. The coherence is computed by picking 1000 random pairs of neurons from the whole extend of the network. Depending on the correlation length of neural activity, pairs of distant neurons could show very low spike-spike coherence and may thus lead to a small overall coherence and obscure local spike-spike coherence. Therefore, the authors should also show the local coherence computed across a local pool of neurons.

Apart from the overall magnitude of coherence, it is unclear whether dense vs. sparse wave regimes could also be distinguished by the presence or absence of a peak in the coherence at the peak frequency of the power spectrum. This difference is suggested by Fig.2i vs. Fig.S1b. In fact, for the dense network the maximal coherence is also quite low (~ 0.06), so the distinguishing feature seems to be the shape of the coherence. Please, clarify the conditions for sparse vs. dense waves more precisely in terms of the magnitude and shape of the coherence function.

Furthermore, the coherence of the full spike trains without eliminating spikes should be shown. Different rates are already accounted for by the normalization of the coherence. The deletion of spikes may introduce some artifacts or artificially reduce the coherence. Especially, I am worried that the deletion fixes spike counts, which completely removes possible effects of spike count variability on the spike-spike coherence. Please also add a mathematical definition of the spike-spike coherence.

Alongside with the coherence, please also show the power spectrum (which is contained in the denominator of the coherence function) in Fig 2. Does it have a band structure (as for the dense wave network in Fig.S1b) or does it lack a peak at a finite frequency (turbulent waves?)?

1c) Another characterization of the sparse-wave regime relies on the comparison of firing rates, CV's and coherence with the corresponding statistics of a non-wave reference network: in the sparse-wave regime these statistics do not change. Although I find this comparison useful for the specific model, I guess it is not useful as a general condition for the sparse-wave regime because it is generally not obvious what would be a suitable reference model. Furthermore, a corresponding comparison of rates, CV's and coherence should also be shown for the dense-wave network and its corresponding random network.

Minor issues

=====

2) Lines 401ff: The discussion on distance-dependent time delays in neural systems needs to be improved. There is extensive literature on the effect of delays on pattern formation, in general, and on traveling waves, in particular, see also the list of some important references at the end. Notably, the existence of traveling waves has been reported in neural fields with distance-

dependent time delays, see, e.g., Atay, Hutt (SIAM, 2004), Venkow et al. (Physica D, 2007) and Veltz (SIAM, 2013), and also in spiking networks with fixed delay, see Roxin and co-workers (2005, 2006, 2011). Recently, traveling waves have also been analyzed in sparsely connected networks of LIF neurons with fixed delay (Senk et al., Physical Reviews Research 2020). It thus needs to be better worked out what the sparse-wave model of the manuscript at hand can add to the wealth of literature on the topic.

3) The authors stress "the importance of distance-dependent time delays". While I understand that distance-dependence is biologically more realistic, it is unclear whether similar results can be achieved with fixed delays (either homogeneously or heterogeneously distributed across the network), see in particular Figure 1G in Kriener et al. (Front. Comput. Neurosci. 2013). Please discuss the distance-dependence of the delay.

4) Another important point is to report the main differences between the sparse-wave regime reported in Ref. 54 and the one studied here. It would be helpful to have a comparison with Ref. 54, e.g., with respect to whether the waves—arising due to different spatial connectivity ranges of excitation and inhibition and due to different membrane time scales $\tau_e \neq \tau_i$ —show similar spike-phase coupling to the LFP, and whether the effects of transient stimulations are comparable to the sparse-wave or dense-wave regimes studied in the manuscript at hand.

5) The sparse wave regime reminds me of the phenomenon of sparse synchrony studied in Brunel & Hakim (Neural Comput. 1999) and Brunel & Wang (J Neurophysiol. 2003). It would be interesting to discuss possible differences and similarities of the mechanisms.

6) Relation between Fig.1A,B (corresponding to lines 99-117), which seems to serve as a motivation of the study, and the rest of the paper is unclear to me. In particular, I do not understand the sentence starting in line 119, and there are no simulations of the network model corresponding to Fig.1A,B.

7) I am not convinced of the statement of Fig.1A,B "Spontaneous network fluctuations are of similar magnitude to stimulus-evoked responses." (l. 464). The behavior in Fig.1A,B seems to be expected from the most naive case, in which single trial fluctuations are composed of a stimulus-evoked component (corresponding to fluctuations of the trial-averaged mean LFP) and trial-by-trial variability (noise) equal to the amount of spontaneous fluctuations: For a weak stimulus the stimulus-evoked component becomes small, and thus only the noise remains leading to fluctuations of the same magnitude as the spontaneous fluctuations. In contrast, the fluctuations of the stimulus-evoked response (i.e. the trial-averaged response) is much smaller.

8) l. 106, l. 468-469: is it 200ms or 100ms? There seems to be an inconsistency in different places of the paper.

9) Fig2i: Is coherence measured between only among E (or I) cells, or also between different cell types?

10) Line 157 and throughout the manuscript: There is no consistency in reporting micrometers either as " μm " or as " μM " or even "uM" (Fig.5a) .

11) Line 222: how do you ensure that you are still in the balanced E/I regime?

12) Line 270: I suppose that the few long-range connections create a small-world topography. Is this correct? And if so, please provide proper references to the wide literature on small-world networks in neuroscience.

13) How is the spike-phase coupling defined? Please add a definition.

14) Line 300,303: The authors ask: "What is the conductance state [...] in the sparse-wave network simulation?". They conclude a high (low) conductance state for sparse- (dense-) wave networks indirectly from the membrane potential distribution. This indirect argument is unclear. Why not measure and compare conductances explicitly for sparse- and dense-wave network

simulations?

15) l.317ff, Fig.7d: How is spike-phase coupling (spike-phase modulation) defined? Please add the mathematical definition. The z-scored numbers of the spike-phase coupling in Fig.7d are hard to interpret. Can you report the spike-phase coupling in its natural units, please?

16) l. 325, 332, 336: please define "circular-resultant". Is it the length of the sample-mean circular resultant vector?

17) Line 322: "(1 nS g_e , 10 nS g_i)" -> ($G_e = 1$ nS, $G_i = 10$ nS)

18) Line 335, Fig. S2: why does the random network exhibit a (weak) spike-phase coupling? Shouldn't the histogram be flat? Please, comment.

19) Line 419: "Fig 4B" -> Fig 4D ?

20) Line 671: $g_{\{e,i\}}$ should have an additional neuron index as in line 679. Also, the subscript i is used for denoting both the index and the inhibitory cell type. Please use different symbols.

21) Line 678: is there also an absolute refractory period after resetting (in line with the experimental findings reported in line 633)?

22) Line 679: $w_{\{e,i\}}$ are unused throughout the manuscript. I suppose they should read $G_{\{e,i\}}$, which are studied in Fig. 3 and reported in Tables S1,2. In the table, the units [S] are missing. Check also the inconsistency of these parameters in lines 763ff: G_e and G_i instead of g_e and g_i ?

23) Line 679: How is the delay treated here? I think the rule in 679 only applies after a time delay.

24) Line 688: What are the explicit values of $\tau_{\{e,i\}}$ used in the simulations? Inhibition faster than excitation?

25) Line 692: How are the excitatory and inhibitory neurons arranged? Along a grid? Are excitatory and inhibitory neurons drawn randomly at each grid point or does the arrangement follow some proper rules? Please, describe the spatial arrangement of the neuronal network in more detail. Maybe provide a mathematical expression for the position of a given neuron j of type $\{e,i\}$.

26) Line 706: What is the physical motivation behind choosing periodic boundary conditions (except for mathematical convenience)? What is the influence of the boundary conditions on the wave patterns?

27) Line 701: what is the value of τ_s in the different simulations? This needs to be explained more carefully.

28) Line 743: "Coherence was then computed [...], and normalized by the pooled power spectrum". This sounds strange: by definition, the coherence contains already a normalization by the power spectra. Please provide a mathematical formula for the coherence.

29) Line 756: For reproducibility, please also add the Brian2 simulation code on the github page.

30) Line 786: What exactly is the value of m ? I suspect $m=80$? Then the second sum should be from 1 to 20? Is it correct that in such a 10x10 pool, the synaptic input current involves the activity of around 300,000 neurons (= 100neurons x 3,000synapses/neuron)?

31) Line 812: gradient "approximated by the appropriate forward and centered finite differences":

Please provide explicit mathematical formulas for the computation of the gradient.

32) Line 813: "phase derivatives were implemented as multiplications in the complex plane". Please, provide formulas. Otherwise it is difficult to understand.

33) line 816 "Wave speed was computed as the ratio of instantaneous frequency to phase gradient magnitude": how is the instantaneous frequency calculated?

34) Figure 1: a,b) How to interpret the plots on the right? In particular, what is the x-axis? What is relative power (is it relative variances?)? I guess the mean is not subtracted when computing variances over time, right? Please provide formulas!

c) why a 9x9 grid? d) at which point in space is the LFP computed? Is it an average over the 10x10 neuron pool, or does it correspond to the LFP at the center neuron of this pool?

e) It needs to be made more explicit what aligned vs unaligned means.

35) Figure 2: how to interpret the raster plots? Is a particular coordinate fixed, say $y = y_0$, and then the y-axis of the raster plot corresponds to the x-coordinate of the plane?

Is the mean rate the mean of all neurons or only of the excitatory ones?

Why are mean rate and LFP z-scored?

36) Figure 4: a) Which quantity is exactly plotted here? Please provide mathematical definition of "Fraction of Wave State"! I guess it is the vertical distance between the blue and the red curve at the dashed line in Fig.3d, right?

37) Figure S1: there is no panel c).

Traveling wave literature (selection):

=====

- Bressloff P. Traveling waves and pulses in a one-dimensional network of excitable integrate-and-fire neurons, *J. Math. Biol.* (2000) 40: 169 – 198.
- Golomb D, Ermentrout GB. Continuous and lurching traveling pulses in neuronal networks with delay and spatially decaying connectivity, *PNAS* (1999) 96(23), 13480-13485.
- Pinto DJ, Ermentrout GB. Spatially structured activity in synaptically coupled neuronal networks: I. Traveling fronts and pulses. *SIAM journal on Applied Mathematics.* 2001;62(1):206-25.
- Liley DT, Cadusch PJ, Dafilis MP. A spatially continuous mean field theory of electrocortical activity. *Network: Computation in Neural Systems.* 2002 Jan 1;13(1):67-113.
- Coombes S, Lord GJ, Owen MR. Waves and bumps in neuronal networks with axo-dendritic synaptic interactions. *Physica D: Nonlinear Phenomena.* 2003 Apr 15;178(3-4):219-41.
- Atay FM, Hutt A. Stability and bifurcations in neural fields with finite propagation speed and general connectivity. *SIAM Journal on Applied Mathematics.* 2004;65(2):644-66.
- Roxin A, Brunel N, Hansel D. Role of delays in shaping spatiotemporal dynamics of neuronal activity in large networks. *Physical review letters.* 2005 Jun 16;94(23):238103.
- Roxin A, Brunel N, Hansel D. Rate models with delays and the dynamics of large networks of spiking neurons. *Progress of Theoretical Physics Supplement.* 2006 Jan 1;161:68-85.
- Venkov NA, Coombes S, Matthews PC: Dynamic instabilities in scalar neural field equations with space-dependent delays. *Physica D* 2007, 232: 1–15.
- Roxin A, Montbrió E. How effective delays shape oscillatory dynamics in neuronal networks.

Physica D: Nonlinear Phenomena. 2011 Feb 1;240(3):323-45.

- Veltz, R. Interplay Between Synaptic Delays and Propagation Delays in Neural Field Equations. SIAM Journal on Applied Dynamical Systems 2013, 12(3), 1566–1612.
- Meijer, H.G., Coombes, S. Travelling waves in models of neural tissue: from localised structures to periodic waves. EPJ Nonlinear Biomed Phys 2, 3 (2014)
- Kriener B, Helias M, Rotter S, Diesmann M, Einevoll GT. How pattern formation in ring networks of excitatory and inhibitory spiking neurons depends on the input current regime. Frontiers in computational neuroscience. 2014 Jan 7;7:187.
- Visser S, Nicks R, Faugeras O, Coombes S. Standing and travelling waves in a spherical brain model: the Nunez model revisited. Physica D: Nonlinear Phenomena. 2017 15;349:27-45.
- Senk J, Korvasová K, Schuecker J, Hagen E, Tetzlaff T, Diesmann M, Helias M. Conditions for wave trains in spiking neural networks. Physical Review Research. 2020 14;2(2):023174.

Reviewer #3 (Remarks to the Author):

Spontaneous traveling waves naturally emerge from horizontal fiber time delays and travel through locally asynchronous-irregular states

In this manuscript, the authors describe a computational study aimed at explaining observed traveling waves and their impact on the network response to stimuli.

The authors cite previous experimental work that observed traveling waves in cortex using extracellular electrode recordings in the awake behaving marmoset. They pose the question of whether the observed traveling waves might be compatible with asynchronous-irregular states, which have been hypothesized to occur in vivo.

To address this question, the authors construct a large network of integrate-and-fire model neurons synaptically connected in such a manner as to generate asynchronous-irregular firing. They observe some traveling wave characteristics and observe an impact on the response to stimuli.

The paper is clearly organized and written. The figures are fine.

In general, I find it mostly unconvincing. Numerous papers have explored AI dynamics and its not surprising that some traveling wave like dynamics can emerge. The fact that fluctuations in hyperpolarization/depolarization state modulates the response to stimuli has also been well characterized previously.

However, my major concern with this paper is the strong assumption (given strong evidence to the contrary) that intrinsically generated asynchronous-irregular states are a dominant mode of cortex during wakefulness and that they could explain traveling waves.

Specifically, the authors should reconcile their findings in the context of two strong experimental results that show that cortex becomes quite silent (within ~10 ms) when thalamus is inactivated:

Guo et al., 2017. Nature Vol 545, p181

Reinhold et al., 2015. Nature Neuroscience. Vol 18. No 12.

Both of these papers provide strong evidence that refutes the hypothesis that cortical activity is intrinsically driven in a sustained manner by such AI dynamics.

Response to Reviewers:

We would like to thank the Editor and all three Reviewers for their thoughtful consideration of the merits of our manuscript: "*Spontaneous traveling waves naturally emerge from horizontal fiber time delays and travel through locally asynchronous-irregular states.*" The Reviewers were very helpful in their reviews, pointing out several points where we were unclear, and highlighting several points where more detail was needed to understand some of the analyses. We have taken their comments to heart and worked hard to address them. In the course of carrying out these new analyses, we discovered a minor bug in our simulation code that resulted in a scaling difference in the calculation of delays across the x and y dimensions of our networks. We have regenerated our datasets and repeated our analysis on the corrected simulations. Our main results and conclusions are unchanged, but we feel it is important to make the Editor and Reviewers aware that each figure panel now displays qualitatively similar but quantitatively different data from the panels in the original submission. Beyond this correction, we have also made several revisions to the manuscript as suggested by the Reviewers, have carried out the suggested analyses, all of which support our original findings, and have made corrections and clarifications where appropriate. Our efforts to address the Reviewers' suggestions and questions have strengthened the manuscript. A detailed list of the specific points we have addressed appears below. Reviewer comments appear in black and our responses appear in blue:

Reviewers' Comments

Reviewer #1

This is an interesting paper in which the authors simulate a large population of LIF neurons that are connected in a topological manner with distance dependent axonal delays. They aim to show that they can maintain irregular activity while at the same time generate traveling waves in the LFP. Finally they show that during certain phases of the wave, there is greater sensitivity to stimuli. The motive of the paper was an earlier paper by the same authors that showed such behavior in marmosets. I have a number of questions about the model and the interpretation. I have no major points but many minor ones that may take some work.

--- Page 6 ---

Authors say:

"While one might expect the organized bands of spiking activity would result in increased correlations across neurons, we found no evidence that this was the case. The introduction of topographic connections"

This seems to contradict the recent paper by Huang et al (Huang, Chengcheng, et al. "Circuit models of low-dimensional shared variability in cortical networks." Neuron 101.2 (2019): 337-348) which showed a low dimensional shared variability both in experiments and in simulations. Do the authors have some idea why this differs from the present work? I believe their network was also sparsely connected as it was in the balanced regime and there were wavelike patterns seen as well.

We thank the Reviewer for bringing up this important point, and appreciate the opportunity to discuss the connections between Huang *et al.* and our manuscript. We do not see the two models as mutually contradictory. Rather, they are intended to explain two different phenomena that occupy distinct frequency ranges. The elegant work of Huang and colleagues offers a mechanistic explanation for a source of shared variability that occurs in sensory cortex and its reduction by attention. As Cohen (a co-author on the Huang study) and Maunsell (2009) showed, this shared variability is strongly reduced when attention is deployed to a stimulus. Related work from our laboratory (Mitchell *et al.*, *Neuron*, 2009) also examined the effects of attention on shared variability and found that the attention-dependent reduction in shared variability is limited to a low frequency range, predominantly below 5 Hz. The Huang model measures correlations within a window of 200 ms, nicely matching the low frequency range.

The model described in our manuscript explains a different phenomenon: our recent discovery that broadband fluctuations in a higher frequency range (above 5 Hz) that (1) are organized into spontaneous traveling waves and (2) at certain phases, improve perceptual sensitivity (Davis*, Muller*, *et al.*, *Nature*, 2020). The two models and their resulting spatiotemporal dynamics thus capture two different phenomena that occupy distinct frequency ranges.

As the Reviewer correctly points out, the waves that occur in the model developed by Huang and colleagues induce correlated variability. Their model shows how correlated variability caused by these waves can be reduced by attention. The waves that occur in our large-scale model do not induce correlations. We believe this difference is due to the fact that the waves in their network model strongly regulate spiking activity, yielding what we would describe as “dense” waves consistent with the waves we observe in our smaller-scale model (Fig. S1, shown below). Related work by the groups of Alain Destexhe and Laurent Perrinet (Yger *et al.*, *J Comput Neurosci*, 2011 and Voges and Perrinet, *Front Comput Neurosci*, 2012), found waves in locally connected networks in the range of 10,000 to 50,000 neurons (similar in scale to the Huang *et al.* study) that also induced strong correlations.

Figure S1. Dense spiking waves induce pairwise correlation in a small-scale spiking network. (a) Spike rasters from 10,000 neurons in a small-scale network simulation with random connections and no delays producing asynchronous-irregular spiking dynamics. The mean firing rate and LFP for a single 100 neuron LFP pool is plotted in black and red respectively. (b) Pairwise spike coherence and power spectral density for LFP pools in (a). (c) Spike rasters as in (a), but with topographic connections and transmission delays (0.2 m/s). (d) Coherence and PSD as in (b) but for the network shown in (c). Dense spike participation in waves generates strong coherence at the dominant frequency of fluctuations in the network. (e) There was no difference in the distribution of unit mean firing rates between the random (black) and topographic (red) small-scale networks ($p = 0.07$; KS test). (f) There was a significant reduction in unit C.V between the random and topographic small-scale networks ($p < 1 \times 10^{-10}$; KS test). (g) Comparison of the PSD between the random and topographic small-scale networks.

The model we explore in the present manuscript is substantially larger in scale -- in the range of 500,000 to 1,000,000 neurons, and this is critical because it enables the network to operate in an asynchronous-irregular regime with smaller synaptic conductances than those that are necessary to generate self-sustained activity in smaller networks with less connectivity (Morrison et al., *Neural Computation*, 2005, 2007). As a result of these differences in scale and synaptic conductance, our model exhibits a novel and qualitatively different dynamical state, where waves of sparse modulations in spiking probability propagate through locally asynchronous states in the network without inducing correlations. We believe our computational model, which has leveraged methodological innovations to create one of the first comprehensive numerical studies with neural counts on the scale of a cortical area and provides new insights into the mechanisms that govern cortical dynamics and their impact on information processing.

Reviewer Figure 1. The network model developed by Huang et al. has strong spiking coherence consistent with our definition of dense waves. Our sparse-wave model (purple line, left panel) does not induce strong spiking coherence, whereas our smaller-scale dense wave producing model (black line) has a significant peak in spiking coherence. The right panel shows the spiking coherence calculated from a simulation generated from the code published by Huang et al. corresponding to the model in their Figure 3Aiv. The large change in spiking probability in the Huang model is consistent with our definition of dense waves. Dotted lines denote the 95% confidence interval.

It is hard to believe that there is no correlation for nearby neurons as a wave passes by or is it just that there was such a low probability of a given neurons being involved in the wave?

The Reviewer's intuition is correct. The probability of any neuron participating in a given wave is so low that there is no significant difference in pair-wise spike coherence as compared to the randomly connected network, which is otherwise equivalent but does not generate waves. The probability of any given neuron spiking at any given millisecond is low, and the increase in spike probability due to the peak of a traveling wave is marginal (2.33% increase averaged over all neurons in LFP bins within $\pi/4$ rad of the peak phase of a wave). While this is a small increase in the spike probability for an individual neuron, this subtle modulation across tens of thousands of neurons is sufficient to mediate the propagation of traveling waves. In contrast, in the smaller dense wave network that appears in Figure S1, the baseline spiking probability increases substantially (26.48% at wave peak), resulting in an increase in pair-wise spike coherence. To make this intuition explicit we have added this quantification to the manuscript (page 6 lines 193-202)

However the difference between sparse and dense is interesting. The authors should plot the pairwise correlations of nearby neurons as a function of density of connection. I'd be interested in how this varies and at which point it is nonzero.

We thank the Reviewer for this comment and for the suggestion to investigate how correlations change with connectivity. In our topographic network, connection density falls off with distance, so we measured the change in spike coherence with distance, comparing the sparse and dense networks. We have added this analysis, which appears as Supplemental Figure S2 and is copied below. Pairwise correlations are not driven differently by distance in the sparse-wave network. In contrast, in the dense-wave network, pairwise spiking coherence is negatively correlated with distance (Pearson's $r = -0.72$).

Figure S2. Distance dependence of pairwise spike coherence. (a) The maximum pairwise spike coherence calculated between neuron pools at various distances in the random (black) and topographic (red) large-scale networks in Figure 2. There was no change in spike coherence in either network at any distance. **(b)** Same as (a), but for the small-scale networks in Figure S1. There was a negative correlation with maximum spike coherence and distance in the small-scale topographic model (Pearson's $r = -0.72$).

--- Page 10 ---

"sparsely modulated"

What does sparsely modulated mean?

We should have made this clear and we appreciate the Reviewer raising this point. Sparseness is not a categorical measure. It corresponds to the degree to which the spiking probability of the neurons in a network vary, on average, as a function of the phase of a traveling wave. In our sparse-wave model, spike probability varied, on average, by 2.33% with wave phase. In our dense-wave network, this phase dependent change in spiking probability was much higher, varying by 26.48%.

--- Page 12 ---

Authors say:

"While previous computational models held that the contribution of axonal delays was negligible in their effects on the temporal properties of spiking networks,"

There are a number of older papers that take distance dependent delays into consideration with regard to waves both in neural oscillations and waves. IN all of these the delays matter and lead to dynamics such as traveling waves among others. These should perhaps be discussed. Here is a list (not comprehensive)

Crook, Sharon M., et al. "The role of axonal delay in the synchronization of networks of coupled cortical oscillators." *Journal of computational neuroscience* 4.2 (1997): 161-172.

Atay, Fatihcan M., and Axel Hutt. "Neural fields with distributed transmission speeds and long-range feedback delays." *SIAM Journal on Applied Dynamical Systems* 5.4 (2006): 670-698.

Golomb, David, and G. Bard Ermentrout. "Continuous and lurching traveling pulses in neuronal networks with delay and spatially decaying connectivity." *Proceedings of the National Academy of Sciences* 96.23 (1999): 13480-13485.

Golomb, David, and G. Bard Ermentrout. "Effects of delay on the type and velocity of travelling pulses in neuronal networks with spatially decaying connectivity." *Network: Computation in Neural Systems* 11.3 (2000): 221-246.

Venkov, N. A., Coombes, S., & Matthews, P. C. (2007). Dynamic instabilities in scalar neural field equations with space-dependent delays. *Physica D: Nonlinear Phenomena*, 232(1), 1-15.

We appreciate the Reviewer's advice and have expanded our discussion (page 14, lines 445-460) to include references to these and other papers that have explored how delays impact network dynamics.

--- Page 15 ---

Authors say:

"Spontaneous LFP fluctuations recorded simultaneously from a spatially distributed multielectrode array were structured as traveling waves"

Note (yellow):

I don't see any traveling wave here. I see a transient increase in amplitude but nothing that looks like an actual moving wave. Am I missing something? Does the fact that the waves are "traveling" change the histogram shape in a way that would be different if

these were just synchronized oscillations? That is, I see that the response is stronger at certain phases of the lfp, but I'd expect the same if no traveling was involved. I don't see the role of the motion of the waves in the enhancement of the response. Indeed, it is hard to see how such motion could be relevant to a non moving stimulus. As an example of traveling waves having a computational role to a perception of a stimulus, see the recent paper of Heitmann and Ermentrout in PLoS Comp Bio 2020

We thank the Reviewer for raising this point. The traveling wave can be seen by focusing on the orange region in Figure 1c, which enters the bottom right corner at 8 ms and moves upward and to the left. We have added as supplemental material videos showing waves in our recorded data (Supplemental Video S1), as well as in our network simulations (Supplemental Videos 3-5). In our recently published manuscript (Davis, Muller, et al, *Nature* 2020), we found that traveling waves were specifically predictive of both the magnitude of the spiking response that was evoked by a faint stimulus and the monkey's ability to detect the target. This was not true of comparably large fluctuations in LFP activity when no traveling wave occurred. Thus, while a synchronized oscillation could, in principle, produce the same effect, we find no evidence that they do, *in vivo*.

--- Page 17 ---

The authors seem unaware of the work of the Doiron group on spatially balanced networks. See the recent paper with Huang et al (op cit) fig 3 and 5 for examples of broad traveling waves. See also the papers with Rosenbaum and Doiron in PRX 2014 and Nature Neuroscience 2017

We thank the Reviewer for bringing these articles to our attention. We are familiar with the work of the Doiron group, particularly the Huang et al paper. The mechanisms of waves in our model and in the Doiron group's model are not mutually exclusive, and we suspect may play distinct roles *in vivo*, potentially at different temporal scales. Our model captures the aspects we observed of traveling waves in our cortical recordings. Spatial and temporal scales of inhibition may play a role in other aspects of cortical dynamics we do not study here.

Have you looked at a simpler one d system and the appropriate mean field spatial model?

We thank the Reviewer for bringing up this important computational and technical point. We have studied a one-dimensional version of this model in comprehensive detail. The model, which is illustrated in the panels below, permits very detailed quantification of the delay-induced emergence of travelling wave dynamics. We have added a supplemental figure (Fig. S4) describing the model. We do feel that a two-dimensional model is important when comparing the model to the 2D waves we observe in cortex, so we focus on the 2D case in the main part of the present manuscript.

There is value in developing mean field models of these phenomena and we plan to do so in the future. Our focus, here, was to make a direct link to the measured timing and phase dependence of spiking and LFP activity. We have chosen to focus on a spiking model that explicitly models both spikes (permitting study of single-neuron dynamics) and the synaptic currents (permitting calculation of the LFP).

Figure S4. Delays are necessary for robust traveling waves in 1-D spiking network model. (a) Schematic of 1-D network model. 450,000 neurons were arranged on a ring with topographic connection probabilities and distance dependent delays. (b) 2-D FFT of the spatial (y-axis) and temporal (x-axis) frequencies of activity in the topographic network. The clear spectral line is consistent with waves traveling at 0.2 m/s. (c) No spectral line appears in a similar topographic 1D network without delays. (d) Spike rasters and LFP amplitude (pseudocolor) for the topographic network displays waves moving across space over time in the 1-D topographically connected network with delays. (e) Same as (d), but for the 1-D topographic network without delays. LFP fluctuations do not travel as waves but rather occur synchronously across regions of the network.

--- Page 18 ---

It may be that the reason that you need axonal delays to get the traveling waves is that you assumed that the spatial spread of excitation and inhibition were the same. As the paper does not specify in the parameter tables the time scales of the e and i synapses I could not say if this was also a reason. As seen in the Huang et al paper, it is easy to get wavelike activity without axonal delays when the spatial and temporal time scales are in the right ranges.

The Reviewer is correct that we use the same spatial and temporal spreads of excitation and inhibition. We have updated Table S1 to reflect this. While it is possible to generate waves without delays in other frameworks, we show that topography with delays is sufficient to produce waves

without needing to tune the spatial and temporal scales of excitation and inhibition. We find that when using delay values corresponding to unmyelinated horizontal fibers in cortex this spiking model can parsimoniously recapitulate the dynamics observed in cortex over a very broad parameter range (see Fig 4a-c).

I am puzzled by what is plotted in 3c. The network is two d and what is plotted is one d. Is this a slice or what is the spatial position denoting?

The Reviewer is correct, the plot in 3c is showing a slice through the 2D network over time. White and black pixels indicate times when those locations did and did not show significant wave activity. We have updated our figure caption and description of results to improve clarity.

If the connectivity is random but there are distance dependent delays will the waves appear? If there is topological connectivity but no delays do the waves appear? I only saw simulations that had both features and I'd like to see which are necessary. That is, if the amplitude coupling is independent of distance but the delay does depend on distance matters more than if the delay is fixed and the amplitude varies with distance. It is known (say in the Crook et al paper above) that distance-dependent delays can destabilize the synchronous state. Is that what is happening here? This is why a mean-field and/or 1-dim network would be useful. It is not clear what the underlying mechanism is.

We thank the reviewer for raising this question. We have added a supplemental figure (Figure-S3, copied below) that shows separately the impact of distance-dependent delays and topological connectivity on traveling waves. Here, we have taken the model with both topographic connections and delays (panel a) and either removed propagation delays (panel b) or randomized connectivity (panel c). We have added an analysis to further quantify the presence of traveling waves by measuring the 2-dimensional FFT over space and time. Activity patterns propagating in the network will produce a spectral line at the slope corresponding to the rate of change shared in spatial and temporal frequencies (Cagigal, et al. *Appl. Opt.* 1995). This slope corresponds to the speed of waves traveling in the network. Removing delays (b) results in a network that does have topographically organized activity as in the full model, but these fluctuations do not move across the network over time as a traveling wave, as demonstrated by the lack of a concentrated band in the space-time FFT (b'). Conversely, a network with delays but no topographic connectivity has fractured spatial structure (c), but does exhibit flowing spatiotemporal activity (c'). As expected, the random network with no delays has neither spatially organized wave lengths nor flowing spatiotemporal activity (d, d'). From these results we conclude that, in our model, topography is sufficient for the spatial organization of wavelengths across the network, and delays are sufficient for the flow of activity across this spatial organization. Both topography and delays are jointly necessary for large-scale traveling waves that resemble those we observe in our cortical recordings.

To further explore the impact of delays, as suggested by the Reviewer, we have made the same comparison in a simplified 1-D model (Supplemental Figure-S4, shown above). The scale and properties of this 1-D model are similar as in the 2-D, but adjusted to the topography of a ring instead of a sheet. Sparse waves occur similarly in this 1-D network, and when examined under a 2-D (space-time) FFT, the spatiotemporal power due to waves appears as a band whose slope corresponds to the delays in the network. When we remove the delay, this spectral line is abolished, indicating delays are critical for the presence of traveling waves.

Figure S3. Topographic connections and distance-dependent delays combined are necessary to generate traveling waves. (a) Significant (white) and non-significant (black) wavelength values for each position in a linear slice through a large-scale 2D network simulation with topographic connections and no delays. **(a')** 2-D (space-time) FFT shows a concentration of spectral power corresponding to waves traveling at the velocity corresponding to propagation speeds (0.2 m/s). **(b, b')** Same as in (a), but for a network with topographic connectivity and no delays. Topographic connectivity is sufficient to generate significant spatially organized wavelengths. However, without delays, the spectral power does not concentrate along a joint spatial and temporal frequency band consistent with traveling waves. **(c, c')** Wavelengths and spatiotemporal FFT for a randomly connected network with delays. With random connectivity the network lacks strong spatial organization while delays are sufficient for the spatiotemporal flow of activity. **(d, d')** Wavelengths and spatiotemporal FFT for a randomly connected network without delays. There is no spatial or temporal structure in this network suggestive of any wave activity.

--- Page 19 ---

The smallest wavelength is 5 mm with the average about 6.5 mm so that there isn't even one wave in the whole network, right since it is only 4 or 6 mm?

The Reviewer is correct, the smallest wavelengths were between 4 and 5 mm, which was larger than the width of the network we were studying (4 mm). The wavelength is estimated from the gradient of phase across the array. If the gradient is gradual, the wavelengths are long, and if the gradients are sharp, the wavelengths are narrow. A wavelength greater than 4 mm corresponds to a total phase offset less than 2π across the network.

--- Page 22 ---

7c is just reflecting your definition in which the LFP is just the difference between g_i and g_e so it should be strongly anticorrelated. Am I missing some bigger point?

This is an important point, and we appreciate the opportunity to clarify this in the manuscript. The LFP proxy we use developed by Mazzoni et al. (PLoS Comput Biol, 2015) is the sum of the absolute values of the excitatory and inhibitory synaptic currents. For this reason, we do not expect the LFP to be negatively correlated with the difference between excitatory and inhibitory synaptic conductances *a priori*, and our observation is not a trivial consequence of the LFP definition: as a simplified example, a given LFP value V could be produced by a total current $I = I_e + I_i$ with $I_e > I_i$ or $I_i > I_e$. In the model, we find the latter is the case at relatively positive LFPs and the former at relatively negative. This generates a strong anticorrelation between the excitatory and inhibitory conductances and the LFP, so that at relatively positive phases, the E-I balance is shifted towards inhibition, while the opposite is true at negative phases (Figure 7c).

We apologize for being unclear in the original submission. While the sign convention used by Mazzoni et al. leaves some ambiguity -- specifically, the excitatory current has a positive sign and the inhibitory has a negative sign, so that the difference $I_e - I_i$ is equivalent to the sum of absolute values $|I_e| + |I_i|$ -- we have worked to revise and clarify this point in the main text (page 11 lines 349-354).

--- Page 24 ---

Fig 8 is the most interesting part of the paper.

We thank the Reviewer for their appreciation of this important finding. We agree the ability of the sparse-wave network to produce gain modulations to weak input, in contrast to the dense-wave network's insensitivity to the same input, is of particular interest.

--- Page 33 ---

In the LFP proxy, where do the values of the parameters come from? That is why is tau 6 and alpha 1.65?

We used the formula from Mazzoni *et al.* (PLoS Comput Biol, 2015), in which the authors inferred the optimal proxy for the LFP in a leaky integrate-and-fire network by validating against a multi-compartmental network model featuring spatially and morphologically realistic neurons. The authors found a robust linear relationship between the underlying leaky integrate-and-fire LFP and the observable excitatory and inhibitory currents (*cf.* Equation [4], page 15 in their paper). Specifically, "tau = 6 ms" is the temporal offset of the excitatory current, and "alpha = 1.65" is the scaling factor of the inhibitory current. The authors found these to best agree with the ground-

truth multi-compartmental model. We appreciate the Reviewer's point and have taken the opportunity to clarify that in the Methods (page 42 lines 987-993).

--- Page 36 ---

"Ermentrout, G. B." is written twice

We have corrected this.

Reviewer #2

The paper presents an extensive simulation study of a large-scale spiking neural network model with the aim to investigate the mechanisms of traveling waves in cortex reported in their previous paper. A mechanistic understanding of cortical traveling waves is of critical importance for understanding the operating regime of cortex. The authors find that distance-dependent transmission delays and connection probabilities in a network of conductance-based integrate-and-fire neurons are sufficient to reproduce the wave activity. They uncover a new regime, where waves co-exist with asynchronous irregular dynamics typical for cortex. In particular, this regime exhibits a flat spike-spike coherence of low magnitude and wave patterns at the same time. The characterization of the waves relies on their previously developed method and a comparison of the wave length' to wave length' extracted from a corresponding random reference network. They call this new regime a sparse-wave regime.

Importantly, the results are compared to experimental data from their previous paper. It is shown that both the spontaneous and evoked activity in this sparse-wave regime is in impressive quantitative agreement with the experimental data. Thus, the major conclusion is that cortical waves operate in the sparse-wave regime. I found the results of the paper very interesting, highly relevant and novel, especially the sparse-wave regime. The simulation and the analysis of simulation data has been performed thoroughly and seems to be technically sound. I think the paper will inspire both experimentalists and theoreticians in neuroscience. My only major concern is the proper characterization of the sparse-wave regime and the computation of the spike-spike coherence. Addressing this issue is important because the sparse-wave regime seems to be the main novelty of the paper. Below I also provide a list of minor issues. In summary, I recommend publication in Nature Communications after these issues have been properly addressed in a revised manuscript.

Major issues

=====

1) The main phenomenon described in this paper is the sparse-wave regime. However, a clear mathematical definition or general conditions for classifying traveling waves as sparse or dense are missing.

1a) Sparse waves are roughly defined by the authors as waves "with only a small fraction of neurons spiking in any given wave". This definition should be made mathematically precise by defining a measure that directly quantifies the notion of the "fraction of neurons spiking in any given wave". E.g one could compute the mean number of spikes "per wave" and neuron, maybe by computing the ratio of mean interspike interval to mean wave duration (at some

local position). This direct measure should be applied to network simulations to convince the reader that the fraction of neurons spiking per wave is indeed much smaller than unity for the presumed sparse-wave network, whereas such a property would not hold for the presumed dense-wave network.

We thank the Reviewer for the very helpful suggestion that we formally define the notions of sparse and dense waves. As suggested, we have calculated the mean number of spikes that occur “per wave” across the population of neurons participating in that wave and related this value to the mean number of spikes observed in total. Using this information we have calculated the change in spiking probability during the excitatory peak of a traveling wave in both the sparse- and dense-wave networks. In the sparse-wave network, units are 2.33% more likely to spike during the peak of a wave whereas in the dense-wave network neurons are 26.48% more likely to spike. Therefore in sparse waves, neurons are weakly modulated in their spiking probability, and in dense waves, neurons are strongly modulated. This is a relational definition, as sparseness and denseness exists on a spectrum, and the boundary between the two is somewhat arbitrary. Functionally, the modulation of spiking probability by sparse waves is weak enough to not result in a change in spiking coherence, whereas the strong modulation of spiking probability by dense waves does induce a significant increase in spiking coherence.

1b) As an indirect way to quantify sparse waves the authors consider the spike-spike coherence and argue that "If it were the case that neurons strongly participated in these fluctuations, then they would show a degree of coherence in the range of frequencies dominated by those fluctuations." While I very much like this idea to use the coherence to characterize sparse waves, I have some concerns about the specific procedure used in the paper. The coherence is computed by picking 1000 random pairs of neurons from the whole extend of the network. Depending on the correlation length of neural activity, pairs of distant neurons could show very low spike-spike coherence and may thus lead to a small overall coherence and obscure local spike-spike coherence. Therefore, the authors should also show the local coherence computed across a local pool of neurons.

The Reviewer raises a valid point regarding the way in which we calculated spike coherence. Instead of using random pairs as previously described, we instead took the 100 excitatory neurons within a LFP pool and calculated the spiking coherence with adjacent LFP pools. Additionally, we measured the maximum coherence as a function of distance between LFP pools across the network. We find there is weak coherence in the large scale network models that does not change as a function of distance, nor does it differ between the random and topographically connected networks, consistent with our previous analysis. Also, we find that the small-scale topographically connected network has significantly stronger local coherence, and this coherence falls off with distance. We have added these results as a supplemental figure (Supplemental Figure S2).

Figure S2. Distance dependence of pairwise spike coherence. (a) The maximum pairwise spike coherence calculated between neuron pools at various distances in the random (black) and topographic (red) large-scale networks in Figure 2. There was no change in spike coherence in either network at any distance. (b) Same as (a), but for the small-scale networks in Figure S1. There was a negative correlation with maximum spike coherence and distance in the small-scale topographic model.

Apart from the overall magnitude of coherence, it is unclear whether dense vs. sparse wave regimes could also be distinguished by the presence or absence of a peak in the coherence at the peak frequency of the power spectrum. This difference is suggested by Fig.2i vs. Fig.S1b. In fact, for the dense network the maximal coherence is also quite low (~ 0.06), so the distinguishing feature seems to be the shape of the coherence. Please, clarify the conditions for sparse vs. dense waves more precisely in terms of the magnitude and shape of the coherence function.

Furthermore, the coherence of the full spike trains without eliminating spikes should be shown. Different rates are already accounted for by the normalization of the coherence. The deletion of spikes may introduce some artifacts or artificially reduce the coherence. Especially, I am worried that the deletion fixes spike counts, which completely removes possible effects of spike count variability on the spike-spike coherence. Please also add a mathematical definition of the spike-spike coherence.

The Reviewer's concern about the rate matching in the coherence analysis is well taken. We have repeated the analysis without the rate matching and confirmed it did not impact our results. We therefore present the results without this unnecessary step. We have also added our mathematical definition of spike-spike coherence to the Methods.

Alongside with the coherence, please also show the power spectrum (which is contained in the denominator of the coherence function) in Fig 2. Does it have a band structure (as for the dense wave network in Fig.S1b) or does it lack a peak at a finite frequency (turbulent waves)?

We have added the power spectrum for each network in Figure 2j. The power spectrum does not have a peak, but rather a broad band consistent with fluctuations at drifting non-stationary frequencies similar to those observed from awake *in vivo* recordings.

1c) Another characterization of the sparse-wave regime relies on the comparison of firing rates, CV's and coherence with the corresponding statistics of a non-wave reference network: in the sparse-wave regime these statistics do not change. Although I find this comparison useful for the specific model, I guess it is not useful as a general condition for the sparse-wave regime because it is generally not obvious what would be a suitable reference model. Furthermore, a corresponding comparison of rates, CV's and coherence should also be shown for the dense-wave network and its corresponding random network.

We would like to thank the Reviewer for their suggestion to compare the dense-wave network spiking properties with its corresponding randomly connected network. This is an important comparison which we have added to Supplemental Figure S1.

Minor issues

=====

2) Lines 401ff: The discussion on distance-dependent time delays in neural systems needs to be improved. There is extensive literature on the effect of delays on pattern formation, in general, and on traveling waves, in particular, see also the list of some important references at the end. Notably, the existence of traveling waves has been reported in neural fields with distance-dependent time delays, see, e.g., Atay, Hutt (SIAM, 2004), Venkow et al. (Physica D, 2007) and Veltz (SIAM, 2013), and also in spiking networks with fixed delay, see Roxin and co-workers (2005, 2006, 2011). Recently, traveling waves have also been analyzed in sparsely connected networks of LIF neurons with fixed delay (Senk et al., Physical Reviews Research 2020). It thus needs to be better worked out what the sparse-wave model of the manuscript at hand can add to the wealth of literature on the topic.

We agree with the Reviewer that we could elaborate our discussion on the role of time delays (both fixed and distance-dependent) in neural systems. We have revised our discussion to refer to the extensive literature on pattern formation in neural networks and how our sparse-wave model adds to that field.

3) The authors stress "the importance of distance-dependent time delays". While I understand that distance-dependence is biologically more realistic, it is unclear whether similar results can be achieved with fixed delays (either homogeneously or heterogeneously distributed across the network), see in particular Figure 1G in Kriener et al. (Front. Comput. Neurosci. 2013). Please discuss the distance-dependence of the delay.

We thank the Reviewer for suggesting we explore the importance of delays in generating traveling waves in our network model. We have generated additional simulations with topographic connectivity and no delays and random connectivity with delays in both our 2-d model, and also a simplified 1-d model. In order to better capture the presence of waves in our networks, we have added the calculation of the space-time FFT (described in our Methods), which reveals a characteristic spectral line in networks with traveling waves, the slope of which corresponds to the propagation speed of waves in the network. In conjunction with the wavelength measure, which quantifies the spatial structure necessary for waves, we can attribute the key components of spatial organization and spatiotemporal flow of activity to topographic connectivity and conduction delays, respectively. Networks with only topographic connectivity have spatial structure, but lack the spatiotemporal band of power indicative of traveling waves. Conversely, networks with delays but random connectivity have spatiotemporal power, but lack coherent spatial structure. These results are even more clear in the 1-d model which simplifies the spatiotemporal paths over which waves can flow. We have added these analyses in supplemental figures (Fig. S3 and S4).

Figure S3. Topographic connections and distance-dependent delays combined are necessary to generate traveling waves. (a) Significant (white) and non-significant (black) wavelength values for each position in a linear slice through a large-scale 2D network simulation with topographic connections and no delays. (a') 2-D (space-time) FFT shows a concentration of spectral power corresponding to waves traveling at the velocity corresponding to propagation speeds (0.2 m/s). (b, b') Same as in (a), but for a network with topographic connectivity and no delays. Topographic connectivity is sufficient to generate significant spatially organized wavelengths. However, without delays, the spectral power does not concentrate along a joint spatial and temporal frequency band consistent with traveling waves. (c, c') Wavelengths and spatiotemporal FFT for a randomly connected network with delays. With random connectivity the network lacks strong spatial organization while delays are sufficient for the spatiotemporal flow of activity. (d, d'). Wavelengths and spatiotemporal FFT for a randomly connected network without delays. There is no spatial or temporal structure in this network suggestive of any wave activity.

Figure S4. Delays are necessary for robust traveling waves in 1-D spiking network model. (a) Schematic of 1-D network model. 450,000 neurons were arranged on a ring with topographic connection probabilities and distance dependent delays. (b) 2-D FFT of the spatial (y-axis) and temporal (x-axis) frequencies of activity in the topographic network. The clear spectral line is consistent with waves traveling at 0.2 m/s. (c) No spectral line appears in a similar topographic 1D network without delays. (d) Spike rasters and LFP amplitude (pseudocolor) for the topographic network displays waves moving across space over time in the 1-D topographically connected network with delays. (e) Same as (d), but for the 1-D topographic network without delays. LFP fluctuations do not travel as waves but rather occur synchronously across regions of the network.

4) Another important point is to report the main differences between the sparse-wave regime reported in Ref. 54 and the one studied here. It would be helpful to have a comparison with Ref. 54, e.g., with respect to whether the waves—arising due to different spatial connectivity ranges of excitation and inhibition and due to different membrane time scales $\tau_e \neq \tau_i$ —show similar spike-phase coupling to the LFP, and whether the effects of transient stimulations are comparable to the sparse-wave or dense-wave regimes studied in the manuscript at hand.

We would like to thank the Reviewer for their suggestion to compare the simulations in Huang et al, 2019 with our network simulations. Importantly, they do not report a “sparse-wave regime” in their network model as this is a unique discovery specific to our work. Unfortunately we are unable

to estimate LFP in their model to measure spike-phase coupling as their code does not permit the output of the necessary excitatory and inhibitory currents for each neuron at each moment (“too much memory” error when running their MATLAB/MEX code to output all currents). However, we did calculate the spike-spike coherence in their network exactly as we did in our own simulations (see figure below, right panel). While in their paper they report relatively weak correlations using 200 ms counting bins, we find, using the same data used in their Figure 3Aiv, very strong pairwise coherence, particularly at 20-30 Hz, which is consistent with the dense-spiking wave regime we observe in the smaller version of our network model (left panel, black line).

Reviewer Figure 1. The network model developed by Huang et al. has strong spiking coherence consistent with our definition of dense waves. Our sparse-wave model (purple line, left panel) does not induce strong spiking coherence, whereas our smaller-scale dense wave producing model (black line) has a significant peak in spiking coherence. The right panel shows the spiking coherence calculated from a simulation generated from the code published by Huang et al. corresponding to the model in their Figure 3Aiv. The large change in spiking probability in the Huang model is consistent with our definition of dense waves. Dotted lines denote the 95% confidence interval.

5) The sparse wave regime reminds me of the phenomenon of sparse synchrony studied in Brunel & Hakim (*Neural Comput.* 1999) and Brunel & Wang (*J Neurophysiol.* 2003). It would be interesting to discuss possible differences and similarities of the mechanisms.

We thank the Reviewer for bringing up this excellent point. In this work, we show that waves traveling over the network can be consistent with the sparse synchrony studied by Brunel and Hakim (*Neural Computation*, 1999) and Brunel and Wang (*J Neurophys*, 2003). The key difference in our model is that these previous studies focus on narrowband oscillatory states - high-frequency 180 Hz activity in Brunel and Hakim (1999) and gamma-band 30-50 Hz activity in Brunel and Wang (2003) - while our work focuses on traveling waves in the irregular fluctuations of the ongoing state of “decorrelated”, asynchronous activity in cortex. We have included a discussion on the content of these papers on page 15 lines (475-479).

6) Relation between Fig.1A,B (corresponding to lines 99-117), which seems to serve as a motivation of the study, and the rest of the paper is unclear to me. In particular, I do not understand the sentence starting in line 119, and there are no simulations of the network model corresponding to Fig.1A,B.

The Reviewer is correct. Fig. 1a and 1b are a motivation for the study of spontaneous fluctuations in cortical activity which we find in our cortical recordings are often structured as traveling waves. While spiking responses are very different between spontaneous and evoked activity, the synaptic drive (estimated from fluctuations in the LFP) is roughly equal between spontaneous and evoked

periods (Fiser et al., *Nature*, 2004). We have clarified this important point. Recently we discovered spontaneous traveling cortical waves are a major contributor to trial-by-trial variability and perceptual sensitivity (Davis, Muller et al. *Nature* 2020). The data presented in the figure are evidence of the phenomenon *in vivo*, and the subsequent simulations are a direct attempt to model the mechanisms that generate their observed properties in large-scale spiking network models. Specifically, the simulations corresponding to Figures 2-7 attempt to model the observed *in vivo* spontaneous activity, and in Figure 8, we offer a glimpse into how this model of spontaneous cortical activity might influence evoked activity similarly to *in vivo*.

7) I am not convinced of the statement of Fig.1A,B "Spontaneous network fluctuations are of similar magnitude to stimulus-evoked responses." (l. 464). The behavior in Fig.1A,B seems to be expected from the most naive case, in which single trial fluctuations are composed of a stimulus-evoked component (corresponding to fluctuations of the trial-averaged mean LFP) and trial-by-trial variability (noise) equal to the amount of spontaneous fluctuations: For a weak stimulus the stimulus-evoked component becomes small, and thus only the noise remains leading to fluctuations of the same magnitude as the spontaneous fluctuations. In contrast, the fluctuations of the stimulus-evoked response (i.e. the trial-averaged response) is much smaller.

The Reviewer has nicely expressed our point that the trial-by-trial variability is equal to the amount of spontaneous fluctuations, as the synaptic drive during spontaneous activity and evoked activity is roughly on parity (Fiser et al., *Nature*, 2004). This motivates the study of spontaneous fluctuations in cortical activity as the state of spontaneous fluctuations are critical in modulating the magnitude of the total evoked response. The spontaneous fluctuations we observe are often structured as waves (Figure 1c), and they are informative about the magnitude of evoked responses (Figure 1d) and the perceptual sensitivity of animals performing a detection task (Figure 1e).

8) l. 106, l. 468-469: is it 200ms or 100ms? There seems to be an inconsistency in different places of the paper.

We thank the Reviewer for catching this error. We have revised the manuscript to be internally consistent with the correct value (200 ms).

9) Fig2i: Is coherence measured between only among E (or I) cells, or also between different cell types?

Coherence is only measured among excitatory cells. We have updated our methods to clarify this important point.

10) Line 157 and throughout the manuscript: There is no consistency in reporting micrometers either as " μm " or as " μM " or even "uM" (Fig.5a) .

Thank you for spotting this inconsistency. All abbreviations of micrometers are now " μm ".

11) Line 222: how do you ensure that you are still in the balanced E/I regime?

We did not by design restrict the simulation parameters to the balanced E/I regime, but rather found which combinations of excitatory and inhibitory synaptic conductance resulted in a network

with balanced, self-sustained activity. Of the 2,500 networks we simulated, 599 produced self-sustained activity consistent with the balanced E/I regime. Our definition is detailed in the subsection "Self-sustained activity" of Methods.

12) Line 270: I suppose that the few long-range connections create a small-world topography. Is this correct? And if so, please provide proper references to the wide literature on small-world networks in neuroscience.

This is an interesting and important point. Because of the large scale of this network (450,000 nodes) and connectivity (3,000), calculating the clustering coefficient and average path length that constitute the small-worldness index (Humphries and Gurney, *PLoS One*, 2008) poses a non-trivial computational effort. Out of interest raised by this question, we are currently working to develop extensions of our custom simulation environment for quantifying network structure in efficient C code. With this in mind, we aim to focus on this quantification in future work.

13) How is the spike-phase coupling defined? Please add a definition.

Spike-phase coupling is defined as the length of the mean resultant vector of the spike-phase distribution. We have added the spike-phase coupling definition to the methods.

14) Line 300,303: The authors ask: "What is the conductance state [...] in the sparse-wave network simulation?". They conclude a high (low) conductance state for sparse- (dense-) wave networks indirectly from the membrane potential distribution. This indirect argument is unclear. Why not measure and compare conductances explicitly for sparse- and dense-wave network simulations?

We thank the Reviewer for this important point. We were mistaken in phrasing this question solely in terms of conductance state. In Figure 7a, our objective was to assess whether the synaptic drive to neurons in the model is clustered and strongly correlated, consistent with a "synchronized" state in cortex, leading to a skewed distribution of membrane potential away from threshold (DeWeese and Zador, *J Neurosci*, 2006), or if it is irregular and uncorrelated, consistent with a "desynchronized" state, leading to a Gaussian distribution of membrane potential just below threshold (Rudolph and Destexhe, *Neural Computation*, 2003). In this way, we meant to focus on the difference in network state between the sparse- and dense-wave models (Destexhe and Contreras, *Science*, 2006; El-Boustani *et al.*, *J Phys Paris*, 2007), rather than specifically on the conductance state. With this in mind, we have revised the main text to focus on this distinction between Gaussian and non-Gaussian membrane potential statistics as a proxy for network state.

We have nonetheless taken the Reviewer's suggestion to heart and analyzed the total conductance of individual neurons selected from the sparse and dense wave models (figure below). Across ten individually selected neurons distributed evenly across each network, total conductances in the sparse wave model had a mean of 227.56 nS, varying as a Gaussian within a relatively narrow range (10th percentile: 166.56 nS, 90th percentile: 294.39 nS), while the dense wave model exhibited periods of very low conductance (10th percentile: 121.53 nS) and very high conductance (90th percentile: 719.72 nS). While we agree with the Reviewer that a complete characterization of the synaptic conductance state of the network is important, we believe it is beyond the scope of the present work and it will be considered in future work.

Reviewer Figure 2. Estimated probability density functions of the combined excitatory and inhibitory conductances (g_e+g_i) for ten neurons in the dense (blue) and sparse (orange) networks.

15) I.317ff, Fig.7d: How is spike-phase coupling (spike-phase modulation) defined? Please add the mathematical definition. The z-scored numbers of the spike-phase coupling in Fig.7d are hard to interpret. Can you report the spike-phase coupling in its natural units, please?

We have added the mathematical definition of spike-phase coupling. The calculation was performed using the `circ_r` function in the Matlab Circular Statistics Toolbox (P. Berens, *CircStat: A Matlab Toolbox for Circular Statistics*, Journal of Statistical Software, Volume 31, Issue 10, 2009), which computes the average phasor modulus, where the arguments of the phasors are the phase values. The values were z-scored as spike-phase coupling is a sum over the number of spikes, and therefore dependent on the mean firing rate (i.e. smaller spike counts are biased toward higher coupling values). However, after correcting an error in our simulation code we observed a different pattern of spike-phase coupling with E/I synaptic conductance that is only weakly influenced by the z-score normalization. We have, as requested, plotted this revised figure in its natural units to improve the interpretation of the results. Importantly, the magnitude of spike-phase coupling is strongly correlated with the strength of synaptic conductances, indicating that stronger synapses produce more synchronous spiking activity. This reveals the importance of having large-scale networks that can achieve stable self-sustained activity with weak synapses to achieve the sparse-coupling observed in our model and *in vivo*.

Figure 7d. Spike-phase coupling was significant across networks in the asynchronous-irregular regime (black border), and the degree of coupling was correlated with the magnitude of synaptic conductances (N = 599 simulations; Pearson's $r = 0.78 \pm 0.001$, 95% CI).

16) l. 325, 332, 336: please define "circular-resultant". Is it the length of the sample-mean circular resultant vector?

The Reviewer is correct, the circular-resultant is the length of the mean of the circular resultant vector. We have revised our description in the Methods to be more explicit.

17) Line 322: "(1 nS g_e , 10 nS g_i)" -> ($G_e = 1$ nS, $G_i = 10$ nS)

Thank you. This has been updated as suggested.

18) Line 335, Fig. S2: why does the random network exhibit a (weak) spike-phase coupling? Shouldn't the histogram be flat? Please, comment.

The Reviewer has made an important observation. Our expectation was the histogram should be flat as well. Upon closer inspection, this result was due to an error in our calculation whereby we were only measuring the spike-phase coupling of a single neuron from each 100 neuron LFP pool, which severely reduced the number of spikes being counted. We have corrected our measurement to sample from all neurons in each LFP pool, reducing the noise in our estimate. The corrected histogram is shown below and, as expected, reveals a uniform distribution of spiking to LFP phase in the randomly connected network. The corrected figure (Previously Figure S2) now appears as Figure S5.

Figure S5. Randomly-connected spiking network model has weak spike-LFP phase coupling. Histogram showing the fraction of spikes that occurred during each phase of the LFP in the randomly connected network shown in Figure 2a (spike-phase index = 0.03).

19) Line 419: "Fig 4B" -> Fig 4D ?

We thank the Reviewer for noticing this error. The correct panel call out is Fig. 4D.

20) Line 671: $g_{\{e,i\}}$ should have an additional neuron index as in line 679. Also, the subscript i is used for denoting both the index and the inhibitory cell type. Please use different symbols.

Thank you. Notation of neuron index has been updated in the variables for membrane potential and conductance as a bracketed superscript.

21) Line 678: is there also an absolute refractory period after resetting (in line with the experimental findings reported in line 633)?

There is a refractory period of 5 ms. This detail has been included in Methods and Table S1.

22) Line 679: $w_{\{e,i\}}$ are unused throughout the manuscript. I suppose they should read $G_{\{e,i\}}$, which are studied in Fig. 3 and reported in Tables S1,2. In the table, the units [S] are missing. Check also the inconsistency of these parameters in lines 763ff: G_e and G_i instead of g_e and g_i ?

Thank you. $w_{\{e,i\}}$ has been replaced with $G_{\{e,i\}}$ to be consistent with the rest of the paper.

23) Line 679: How is the delay treated here? I think the rule in 679 only applies after a time delay.

That is correct. First is the delay. Then, the postsynaptic membrane potential is updated, and if it exceeds threshold, the spike and reset conditions occur. This has been clarified in Methods.

24) Line 688: What are the explicit values of $\tau_{\{e,i\}}$ used in the simulations? Inhibition faster than excitation?

The time constant for excitation and inhibition was 5 ms in all simulations. This has been added to Table S1.

25) Line 692: How are the excitatory and inhibitory neurons arranged? Along a grid? Are excitatory and inhibitory neurons drawn randomly at each grid point or does the arrangement follow some proper rules? Please, describe the spatial arrangement of the neuronal network in more detail. Maybe provide a mathematical expression for the position of a given neuron j of type $\{e,i\}$.

Each component of the position $(x^{(i)}, y^{(i)})$ of neuron i is randomly drawn from the uniform distribution $U(0, L)$, where L is the side length of the square space over which the network is defined. This position assignment rule applies to all neurons $i \in \{1, 2, 3, \dots, N\}$, where N is the number of excitatory and inhibitory neurons. Thank you for bringing this important point to our attention. It has been clarified in Methods.

26) Line 706: What is the physical motivation behind choosing periodic boundary conditions (except for mathematical convenience)? What is the influence of the boundary conditions on the wave patterns?

Periodic boundary conditions are necessary for the connectivity conditions to be homogeneous across the network. Without periodic boundary conditions, neurons on the edges of the network would either have fewer connections or a higher degree of local connectivity than neurons in the center of the network, potentially leading to inhomogeneities in the activity dynamics across the network.

27) Line 701: what is the value of τ_s in the different simulations? This needs to be explained more carefully.

The value of τ_s is 300 μ s in all simulations. Thus, in the networks lacking distance-dependent delays, τ_s is the only delay modeled, and in the other networks, which contain distance-dependent delays, τ_s is contained in addition. This parameter has been added to Table S1.

28) Line 743: "Coherence was then computed [...], and normalized by the pooled power spectrum". This sounds strange: by definition, the coherence contains already a normalization by the power spectra. Please provide a mathematical formula for the coherence.

We have updated our methods to include the mathematical formula for computing the spike-spike coherence which includes the normalization by the power spectra.

29) Line 756: For reproducibility, please also add the Brian2 simulation code on the github page.

We have added the Brian2 simulation code to the github page as requested.

30) Line 786: What exactly is the value of m ? I suspect $m=80$? Then the second sum should be from 1 to 20? Is it correct that in such a 10x10 pool, the synaptic input current involves the activity of around 300,000 neurons (= 100neurons x 3,000synapses/neuron)?

The value of m is 100 excitatory neurons. In each of the 10x10 pools of neurons used to calculate the LFP, there are 100 excitatory and 20 inhibitory neurons. The LFP is calculated from the excitatory and inhibitory synaptic inputs arriving from neurons within and outside the pool across those 300,000 (100 neurons x 3000 inputs / neuron) inputs.

31) Line 812: gradient "approximated by the appropriate forward and centered finite differences": Please provide explicit mathematical formulas for the computation of the gradient.

32) Line 813: "phase derivatives were implemented as multiplications in the complex plane". Please, provide formulas. Otherwise it is difficult to understand.

33) line 816 "Wave speed was computed as the ratio of instantaneous frequency to phase gradient magnitude": how is the instantaneous frequency calculated?

Thank you for seeking clarification on these important matters. All three mathematical formulas are now included.

34) Figure 1: a,b) How to interpret the plots on the right? In particular, what is the x-axis? What is relative power (is it relative variances?)? I guess the mean is not subtracted when computing variances over time, right? Please provide formulas!

The plots to the right in Figure 1 a, b are hive plots. There is no meaning to the dispersal along the x-axis, but is simply to eliminate overlap for the ease of viewing the points in the distribution. The plots are to be interpreted by their height on the y-axis, with a value near zero indicating the spontaneous and evoked periods had similar fluctuation amplitudes. The more distant the value from zero, the greater the disparity between the amplitude of the spontaneous and evoked LFP

fluctuations. The relative power is the ratio in power between the LFP signal in the stimulus and evoked periods.

c) why a 9x9 grid? d) at which point in space is the LFP computed? Is it an average over the 10x10 neuron pool, or does it correspond to the LFP at the center neuron of this pool?

We apologize to the Reviewer for not making the nature of this figure clearer. Figure 1c is from *in vivo* multielectrode array recordings. The 9x9 pixels correspond to electrode positions on a physical recording array implanted in area MT of marmoset cortex. The colors indicate the amplitude of the LFP recorded at the electrodes. We have clarified our description of this panel in the manuscript.

e) It needs to be made more explicit what aligned vs unaligned means.

In Figure 1e “aligned” refers to when the depolarizing phase of a wave is aligned with the retinotopic location of a visual target in cortical area MT. “Unaligned” refers to when the opposite phase is aligned with the retinotopic location of a visual target. We have updated our figure description to clarify this ambiguous information.

35) Figure 2: how to interpret the raster plots? Is a particular coordinate fixed, say $y = y_0$, and then the y-axis of the raster plot corresponds to the x-coordinate of the plane? Is the mean rate the mean of all neurons or only of the excitatory ones? Why are mean rate and LFP z-scored?

The y-axis of the raster plot corresponds to 10,000 neurons evenly spaced along a 1-D slice through the network. The mean rate is the rate of the 100 excitatory neurons within a single LFP pool. We have updated this explanation in the text.

36) Figure 4: a) Which quantity is exactly plotted here? Please provide mathematical definition of "Fraction of Wave State"! I guess it is the vertical distance between the blue and the red curve at the dashed line in Fig.3d, right?

The “Fraction of Wave State” is the fraction of LFP pools (at each point in space and time) in the network that are showing significant wave activity out of the total number of pools. That equates to the area between the observed and shuffled wavelengths above the red dashed line.

37) Figure S1: there is no panel c).

We thank the Reviewer for bringing this error to our attention. We have corrected this oversight in our revision.

Traveling wave literature (selection):

=====

- Bressloff P. Traveling waves and pulses in a one-dimensional network of excitable integrate-and-fire neurons, J. Math. Biol. (2000) 40: 169 – 198.
- Golomb D, Ermentrout GB. Continuous and lurching traveling pulses in neuronal networks with delay and spatially decaying connectivity, PNAS (1999) 96(23), 13480-13485.

- Pinto DJ, Ermentrout GB. Spatially structured activity in synaptically coupled neuronal networks: I. Traveling fronts and pulses. *SIAM journal on Applied Mathematics*. 2001;62(1):206-25.
- Liley DT, Cadusch PJ, Dafilis MP. A spatially continuous mean field theory of electrocortical activity. *Network: Computation in Neural Systems*. 2002 Jan 1;13(1):67-113.
- Coombes S, Lord GJ, Owen MR. Waves and bumps in neuronal networks with axo-dendritic synaptic interactions. *Physica D: Nonlinear Phenomena*. 2003 Apr 15;178(3-4):219-41.
- Atay FM, Hutt A. Stability and bifurcations in neural fields with finite propagation speed and general connectivity. *SIAM Journal on Applied Mathematics*. 2004;65(2):644-66.
- Roxin A, Brunel N, Hansel D. Role of delays in shaping spatiotemporal dynamics of neuronal activity in large networks. *Physical review letters*. 2005 Jun 16;94(23):238103.
- Roxin A, Brunel N, Hansel D. Rate models with delays and the dynamics of large networks of spiking neurons. *Progress of Theoretical Physics Supplement*. 2006 Jan 1;161:68-85.
- Venkov NA, Coombes S, Matthews PC: Dynamic instabilities in scalar neural field equations with space-dependent delays. *Physica D* 2007, 232: 1–15.
- Roxin A, Montbrió E. How effective delays shape oscillatory dynamics in neuronal networks. *Physica D: Nonlinear Phenomena*. 2011 Feb 1;240(3):323-45.
- Veltz, R. Interplay Between Synaptic Delays and Propagation Delays in Neural Field Equations. *SIAM Journal on Applied Dynamical Systems* 2013, 12(3), 1566–1612.
- Meijer, H.G., Coombes, S. Travelling waves in models of neural tissue: from localised structures to periodic waves. *EPJ Nonlinear Biomed Phys* 2, 3 (2014)
- Kriener B, Helias M, Rotter S, Diesmann M, Einevoll GT. How pattern formation in ring networks of excitatory and inhibitory spiking neurons depends on the input current regime. *Frontiers in computational neuroscience*. 2014 Jan 7;7:187.
- Visser S, Nicks R, Faugeras O, Coombes S. Standing and travelling waves in a spherical brain model: the Nunez model revisited. *Physica D: Nonlinear Phenomena*. 2017 15;349:27-45.
- Senk J, Korvasová K, Schuecker J, Hagen E, Tetzlaff T, Diesmann M, Helias M. Conditions for wave trains in spiking neural networks. *Physical Review Research*. 2020 14;2(2):023174.

We greatly appreciate this list of references, which we have included in our updated manuscript.

Reviewer #3

Spontaneous traveling waves naturally emerge from horizontal fiber time delays and travel through locally asynchronous-irregular states

In this manuscript, the authors describe a computational study aimed at explaining observed traveling waves and their impact on the network response to stimuli.

The authors cite previous experimental work that observed traveling waves in cortex using extracellular electrode recordings in the awake behaving marmoset. They pose the question of whether the observed traveling waves might be compatible with asynchronous-irregular states, which have been hypothesized to occur in vivo.

To address this question, the authors construct a large network of integrate-and-fire model neurons synaptically connected in such a manner as to generate asynchronous-irregular firing. They observe some traveling wave characteristics and observe an impact on the response to stimuli.

The paper is clearly organized and written. The figures are fine.

In general, I find it mostly unconvincing. Numerous papers have explored AI dynamics and its not surprising that some traveling wave like dynamics can emerge. The fact that fluctuations in hyperpolarization/depolarization state modulates the response to stimuli has also been well characterized previously.

However, my major concern with this paper is the strong assumption (given strong evidence to the contrary) that intrinsically generated asynchronous-irregular states are a dominant mode of cortex during wakefulness and that they could explain traveling waves.

Specifically, the authors should reconcile their findings in the context of two strong experimental results that show that cortex becomes quite silent (within ~10 ms) when thalamus is inactivated:

Guo et al., 2017. Nature Vol 545, p181

Reinhold et al., 2015. Nature Neuroscience. Vol 18. No 12.

Both of these papers provide strong evidence that refutes the hypothesis that cortical activity is intrinsically driven in a sustained manner by such AI dynamics.

We thank the Reviewer for their comments on our manuscript. They spurred our thinking and this has improved the paper. First, with regard to the point the Reviewer raises about novelty, we have edited the manuscript to make the novelty of this work clear. We do agree that prior modeling work has explored A-I dynamics and conceptually it is not surprising that wave-like dynamics may emerge through various mechanisms. The present results, we think, do go beyond the existing literature by introducing a newly discovered emergent property of large-scale topographically organized networks with transmission delays. We show that networks with these properties naturally give rise to traveling waves that are both compatible with A-I dynamics and do not induce correlations in spiking activity. An extensive body of theoretical work has characterized the potentially deleterious effects of correlated variability, and recent work from our laboratory (Nandy et al., 2018) has shown that induction of low frequency correlations does measurably impair

sensory discrimination. In the context of this work, this newly discovered “sparse wave regime”, which is observed only in very large scale single-unit models that far exceed prior models of A-I dynamics, is important because it explains how the brain can gain the benefits of traveling waves (substantial improvements in perceptual sensitivity; Davis et al, 2020) without inducing potentially harmful correlated variability.

We also thank the Reviewer for bringing up the Reinhold, et al. and Guo, et al. papers. Both papers beautifully show that inactivation of thalamic input to the cortex reduces cortical spike rates in the mouse brain, and we imagine that a similar pattern would hold if these experiments were repeated in the primate. We agree that thalamic input may be a necessary condition in order for waves to occur. It may be, for example, that a minimum level of ongoing cortical activity is needed to support waves, and inactivation of thalamus might reduce cortical firing rates below that level. We have done additional work to examine this question, as elaborated further, below.

But first, we do feel we should make clear that our goal in the present work was to develop the simplest model we could conceive of that could generate traveling waves that recapitulate the waves we observe *in vivo*, and to use this model to gain insight into the impact of traveling waves on properties such as cortical response variability. To pursue this goal we adopted the strategy of starting with a single two dimensional network framework that has previously been used to model the spiking dynamics of populations of neurons in cortex -- the elegant and well-studied cortical model that was introduced by Brunel and colleagues (Brunel, *J Comput Neurosci.*, 2000) to study A-I dynamics. We then asked if that model could, with two simple changes (incorporation of distance-dependent connectivity and axonal propagation delays), generate traveling waves similar to those we discovered in the marmoset. The dynamics in our model are well matched to the dynamics we observed *in vivo*. Since the Brunel network gives rise to neuronal populations that operate in the asynchronous-irregular regime, this strategy positioned us to ask if waves disrupt A-I dynamics.

We fully agree with the Reviewer that, *in vivo*, cortical areas receive many sources of external input that undoubtedly impact the generation and properties of spontaneous activity. Our effort was in no way meant to deny this. We certainly do not mean to call into question the importance of thalamo-cortical connectivity. Nor do we mean to challenge the importance of input from other cortical areas, which studies in the primate have shown can have profound effects on both stimulus-evoked and ongoing activity. These topics are of great interest to us, and we already have experiments underway looking at inter-areal interactions and their influence on traveling wave activity. As this work unfolds, we fully intend to modify the present simple model to take these effects into account. We have added a section to the Discussion explaining the thinking that motivated our approach and pointing out that these inputs may play a role.

That said, struck by the Reviewer’s comments, we have taken a first step in modeling the influence of thalamic input on traveling waves, using the Reinhold and Guo findings as a starting point.

Reinhold and colleagues studied the effects of thalamic inactivation in both anesthetized and awake mice. Under anesthesia thalamic silencing had no effect on spontaneous activity. As they state in describing Figure S3b, “Spontaneous activity in V1 is not affected by silencing thalamus” in anesthetized mice. They did, however, find that in awake mice, thalamic inhibition did reduce spontaneous activity. The effect was transient and began to recover after ~125 ms. After this initial suppression, the cortical activity rebounded. As they note “... dLGN silencing in awake animals was invariably followed by a rebound of activity (recorded in both dLGN and V1) approximately 250 ms after laser onset, although the laser remained on. This rebound was the

first cycle of a thalamo-cortical oscillation (4–8 Hz) lasting for a second or more.” Guo et al. found similar patterns in motor-related area ALM, and the effects appear even stronger. Upon inactivation of the relevant thalamic region, cortical spiking activity was profoundly suppressed (Fig 3). As was seen in the visual system, cortical activity recovered to some extent after this initial period (leveling off in the range of ~20-30% of pre-inactivation baseline).

To model this, we created a modified model in which we incorporate surrogate thalamic input and examine its impact on traveling wave activity. Here, we titrated the strengths of intrinsic cortical connections and thalamic input so that (1) in the presence of thalamic input, the mean firing rate approximated that of our original model and (2) the removal of thalamic input roughly approximated the reduction in spontaneous cortical firing rates that were observed by Reinhold et al. and by Guo et al. Thalamic contributions were modeled as random Poisson spiking inputs driving activity throughout the network. As shown in the figure below, removal of thalamic input led to a substantial reduction in ongoing cortical spike rates (panel a). Panel b shows LFP fluctuations during both the period of feed-forward driving input, and after it is abolished, leaving the remaining activity solely driven by intrinsic self-sustained dynamics. During both periods LFP peaks can be seen moving across the network as traveling waves. To validate the presence of traveling waves we applied a 2-D FFT analysis (described in detail in our Methods) which reveals bands of power corresponding to the waves of activity traveling consistently across spatial and temporal dimensions of the network during both the presence (panel c) and the absence (panel d) of feed-forward driving input.

While this model demonstrates waves can also emerge when activity is externally driven rather than intrinsically generated, it is an early effort and we do not feel that it merits inclusion in the paper. First, it does not advance the original goals of the model in the paper, which were, as mentioned above, (1) to test whether a realistically scaled cortical model with distance-dependent connectivity and realistic transmission delays is sufficient to give rise to traveling waves whose dynamics match those of waves in marmoset MT and (2) to test whether traveling waves are compatible with asynchronous-irregular dynamics or instead alter pairwise spiking statistics.

Further, while the external input takes a step toward modeling the role thalamic input may play, a serious answer to this question would be a major project in and of itself, requiring additional experiments involving recordings and causal perturbations within the thalamus and cortex of the marmoset, as well as a comprehensive model that includes not only feedforward thalamic input but also corticothalamic feedback. We do thank the Reviewer, again, for raising these questions. We hope this additional modeling effort, coupled with the additions to the Discussion section (1) clarifying the novelty of the model and (2) clarifying both the intention behind this modeling effort and the potential importance of thalamic and cortical inputs will, to some extent, assuage the Reviewer’s concerns.

Reviewer Figure 3. Waves persist in a topographic spiking network model receiving feed-forward spiking input analogous to thalamocortical driving input. (a) A network simulation where the majority of the spiking activity was driven by feed-forward spiking inputs arriving analogous to a thalamic source. The mean firing rate of the network (~ 4.5 Hz) during feed-forward input was significantly reduced when thalamic input was turned off, with a reduction similar in magnitude to that observed during experimental thalamic inactivation ($\sim 75\%$ reduction). **(b)** LFP amplitude (z-score) from a 1-D slice through the network during the period before and after removal of the driving inputs. Waves can be seen in both conditions as angled peaks in the LFP moving across the spatial extent of the network over time (examples indicated by arrows). **(c)** 2-D FFT during the period of feed-forward driving input reveals a band of spatiotemporal power consistent with the presence of traveling waves. **(d)** Same as in (c), but after feed-forward driving input is abolished. Waves are still present when the reduced network activity is entirely intrinsically driven.

REVIEWERS' COMMENTS

Reviewer #2 (Remarks to the Author):

The authors have addressed all my comments in detail and satisfactorily. Therefore I would like to recommend acceptance of the paper in Nature Communications.

Reviewer #3 (Remarks to the Author):

Thanks to the authors for their robust responses to the review comments. While there clearly remain many open questions about the nature and role of the activity they observe in vivo, they do present an interesting model that replicates the phenomena and recapitulates a novel network state.

My concerns have been addressed.

Attached is our response to the final Reviewer comments, reproduced verbatim below. Reviewer comments are in black, our response is in blue.

REVIEWERS' COMMENTS

Reviewer #2 (Remarks to the Author):

The authors have addressed all my comments in detail and satisfactorily. Therefore I would like to recommend acceptance of the paper in Nature Communications.

We thank the Reviewer for their helpful review of our manuscript and are pleased we have addressed their concerns satisfactorily.

Reviewer #3 (Remarks to the Author):

Thanks to the authors for their robust responses to the review comments. While there clearly remain many open questions about the nature and role of the activity they observe in vivo, they do present an interesting model that replicates the phenomena and recapitulates a novel network state.

My concerns have been addressed.

We agree with the Reviewer that there are many intriguing open questions that we are excited to explore in the future. We are pleased the Reviewer's concerns have been addressed and thank them for their thoughtful consideration of our manuscript.